# Carbonated mantle peridotites represent a hidden sink for subducted CO$_2$

Elliot J. Carter [1,2] ✉, Brian O'Driscoll [3], Ray Burgess [1], Patricia L. Clay [3], Hélène Balcone-Boissard[4], Pierre Bürckel[5] & the Oman Drilling Project Science Team*

Subduction of carbon rich sediments and crust at convergent plate boundaries exerts a crucial control on Earth's mantle chemistry and surface habitability. Recent attention has focused on exposures of fully-carbonated mantle rocks as these may attest to an overlooked sink for subducted carbon not sampled by arc volcanism. However, even in the best-studied example, the Semail Ophiolite, Oman, the setting for carbonation remains highly contentious, with conflicting inferences from geochemistry and geochronology. We approach this problem by combining microanalysis of halogens and detailed petrography to fingerprint the origins of carbonating fluids. Fluids were derived from both sedimentary pore fluid expulsion and deep slab decarbonation reactions in a subduction zone setting. Through mass balance modelling we show that CO$_2$ fluxes into the forearc from deep decarbonation ($1.7-3.4 \times 10^{13}$ gyr$^{-1}$ C) could represent up to 90% of the global flux entering subduction zones, indicating that carbonated mantle peridotites likely represent a major sink for subducted CO$_2$ which may have varied through geological time.

Subduction of altered oceanic lithosphere rich in volatile elements including carbon, water and the halogens, exerts a major control on the long-term evolution of the atmosphere and oceans, and ultimately the habitability of Earth and other planets[1–3]. Establishing the magnitude of volatile fluxes into the deep Earth or of their return to the atmosphere requires evaluating mass balance between inputs to the subduction system – via altered oceanic lithosphere – and outputs – via slab dehydration and arc volcanism[4–6]. Recently attention has turned to the potential for hidden outfluxes not sampled by volcanism and unaccounted for in current estimates[6–10]. Slab-parallel return of subducted CO$_2$ into the forearc mantle represents one such potential hidden flux.

Mantle peridotites react readily with CO$_2$ to form listvenites – fully-carbonated rocks in which silicate phases such as olivine and pyroxene are replaced by magnesite (MgCO$_3$), dolomite (CaMg[CO$_3$]$_2$) and quartz[11]. The net result of listvenite formation is to

bind aqueously-dissolved CO$_2$ into solid, inert minerals which are stable over geological timescales. Their occurrence as natural products of mineral carbonation provide an invaluable opportunity to evaluate the extent to which CO$_2$-metasomatism of the shallow mantle impacts global carbon cycling at subduction zones.

Carbonation reactions at convergent margins have been indirectly implicated by the carbon isotopic composition of forearc spring gases[8]. However, although listvenites are documented in numerous ophiolites worldwide[12–14] there are few whose formation is unambiguously linked to contemporaneous subduction[11]. In the Semail Ophiolite, Oman, arguably the best studied example of a supra-subduction zone ophiolite[15], deposits of listvenite overlie the basal thrust of the ophiolite and are estimated to have naturally sequestered 1 GT of CO$_2$[4].

This study focuses on listvenites from Hole BT1b of the Oman Drilling Project[16] drilled in Wadi Mansah (23.364374°N, 58.182693°E) in the Semail Ophiolite. Listvenites from BT1b and nearby exposures

[1]Department of Earth and Environmental Sciences, University of Manchester, Manchester, UK. [2]School of Life Sciences, Keele University, Newcastle-under-Lyme, UK. [3]Department of Earth and Environmental Sciences, University of Ottawa, Ottawa, Canada. [4]ISTeP, Sorbonne Université, CNRS, Paris, France. [5]Institut de Physique du Globe de Paris (IPGP), CNRS, Université de Paris, Paris, France. *A list of authors and their affiliations appears at the end of the paper. ✉e-mail: e.carter2@keele.ac.uk

show elevated $^{87}Sr/^{86}Sr$ ratios (0.7092–0.7145) relative to uncarbonated peridotites (~0.7028), Cretaceous to modern seawater or groundwater (~0.707–0.709), but which overlap the isotopic composition of underlying allochthonous Hawasina metasediments (0.7082–0.7241)[17,18]. Clumped isotopes indicate carbonate formation temperatures between 50 and 250 °C with most <150 °C, corresponding to inferred depths of ~1–10 km[17,19]. Trace element geochemistry shows pronounced enrichment of fluid mobile elements in listvenites in a pattern mirroring that of forearc serpentinites[20].

However, there appears to be a mismatch between interpretations based on trace element and isotope geochemistry, which both suggest a forearc subduction zone setting for listvenite formation[17,18,20,21], and those from recent in-situ U-Pb geochronology on samples from localities near (<10 km) Hole BT1b[22]. The latter yielded dates from two dolomite veins and several imprecise dates from listvenites, all <60 Myr, postdating active subduction by at least 20 Myr. The dichotomy between these viewpoints is significant: in the former case, listvenite formation is a volumetrically appreciable and potentially ubiquitous element of carbon cycling at subduction zones, in the latter it is a product of some late-stage tectono-magmatic event in Oman's geological history (e.g., elevated heatflow related to minor Paleogene magmatism[23]) and of regional rather than global significance.

In order to resolve this question, we analysed halogen geochemistry in a suite of progressively-carbonated serpentinites and listvenites from the Semail Ophiolite. The heavy halogens (Cl, Br, I) are hydrophilic and make ideal tracers for interrogating fluid processes in ophiolites[24–26]. Combining secondary ionisation mass-spectrometry and neutron-irradiation noble gas mass-spectrometry to analyse all four halogens in situ and in bulk, we are able to fingerprint fluid provenance as well as discriminate between the effects of fluid-mineral partitioning and fluid evolution. We identify two petrographically and chemically distinct carbonation events and suggest the latter of these

likely correlates with young ages from geochronology while the main stage of carbonation has a subduction origin. We show that the halogen compositions of these fluids have been modified by Cl release associated with carbonation. Mass balance modelling of that fluid evolution indicates significant extra input of $CO_2$ transported by lateral flow from zones of decarbonation deeper in the subduction zone is required, suggesting carbonation at the leading edge of the mantle wedge may be a significant and overlooked part of the global carbon and other volatile element cycles (e.g., $H_2O$, Cl).

## Results & discussion
### Geological context and sample selection
The ~96 Ma Semail Ophiolite, Oman, represents the largest known and arguably best-studied ophiolite in the world (Fig. 1a). The basal thrust of the ophiolite tectonically juxtaposes mantle peridotites on top of allochthonous metasediments and metabasalts of the Hawasina and Haybi complexes which are themselves underlain by autochthonous shelf carbonates[27,28]. A greenschist to amphibolite grade metamorphic sole formed by subduction and exhumation of allochthonous lithologies is discontinuously exposed along the basal thrust[28,29]. The mantle section has recently been sampled by diamond core drilling, as part of the Oman Drilling Project[16], yielding an exceptionally well-characterised sample set.

The core from Hole BT1b consists of 300 m of listvenites, serpentinites, ophicarbonates and metasedimentary/metabasaltic rocks with 100% recovery (Fig. 1b). It is divided into an upper and lower listvenite, the latter characterised by several broad zones containing the chromium-rich mica fuchsite and by elevated concentrations of many fluid-mobile trace elements[20]. Within the upper listvenite a 20 m thick lens of serpentinite and ophicarbonate (partially-carbonated serpentinite) is preserved (80–100 m depth) and affords almost complete preservation across a ~5 m thick reaction front between the

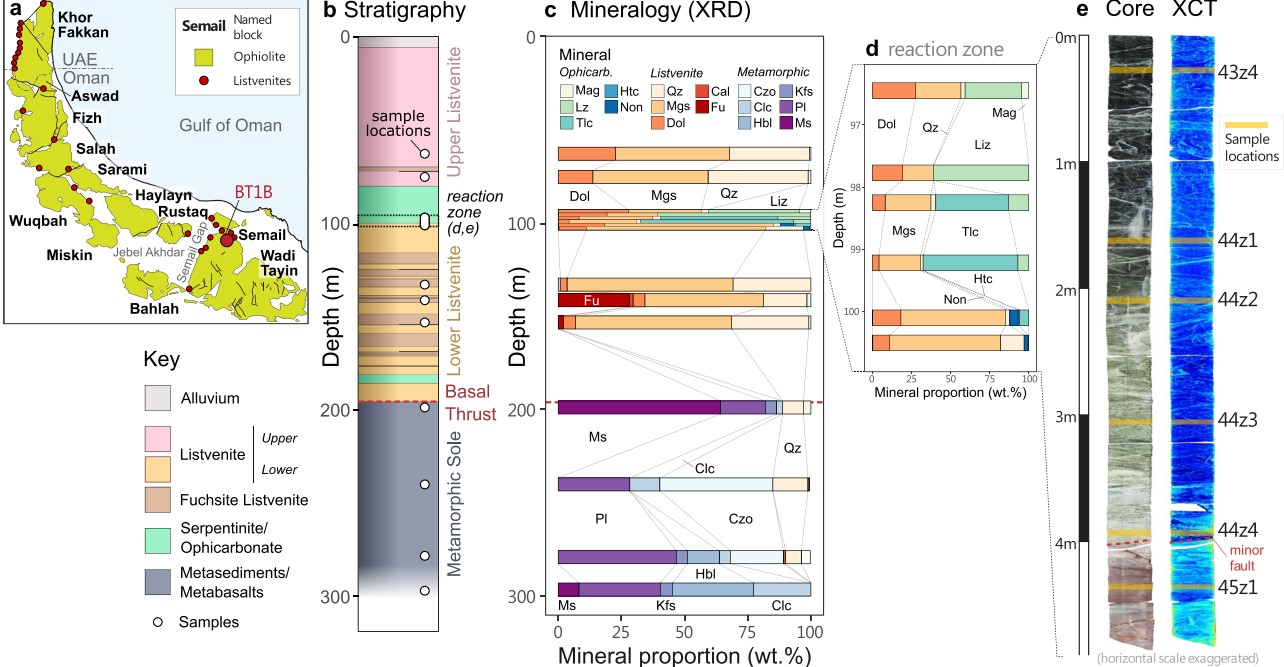

**Fig. 1 | Location, pseudostratigraphy and mineralogy of core BT1b. a** Map of the Semail ophiolite showing location of Hole BT1b and other listvenite localities from the literature[78,79]. **b** Pseudostratigraphic column of Hole BT1b showing sample locations and the relationship between listvenite, serpentinite/ophicarbonate and metamorphic lithologies, the latter underlying the basal thrust fault. The hole was drilled sub-perpendicular to lithological boundaries so depth is approximately true thickness[16]. **c** Proportions of minerals (wt%) in whole rock samples determined by powder XRD **d** inset showing mineralogy of reaction zone samples. **e** Colour

photographs and computed X-Ray tomographic (XCT) sections of the reaction zone between serpentinite and listvenite; sample locations are overlaid in orange. The location of a minor fault removing an unknown thickness of rock between samples 44z4 and 45z1 is shown in red. Mineral abbreviations: Mag magnetite, Lz lizardite, Tlc talc, Htc hydrotalcite, Non nontronite, Qz quartz, Mgs magnesite, Dol dolomite, Cal calcite, Fu fuchsite, Czo clinozoisite, Clc clinochlore, Hbl hornblende, Kfs K-feldspar, Pl plagioclase, Ms muscovite.

serpentinite protolith and fully-carbonated listvenite (Fig. 1e). This interval shows a sharp compositional gradient over which the primary mineralogical variation is due to the degree of interaction with the carbonating fluid.

This study comprises a detailed microanalytical investigation of halogen geochemistry in six samples spaced across this reaction zone, supplemented by whole rock analyses of a broader sample set including representative examples of ophicarbonate, listvenites (± fuchsite) and metamorphic lithologies (Fig. 1).

## Petrography of reaction zone samples

The reaction zone shows a continuous macroscopic progression in colour, from very dark through increasingly-pale green ophicarbonate (samples 43z4, 44z1, 44z2, 44z3) to pale green and then red listvenite (44z4, 45z1) (Fig. 1e).

Petrographic observations and mineralogical analysis by powder X-ray diffraction (XRD) reveal that ophicarbonates consist predominantly of mesh-textured lizardite with variable replacement by carbonate and talc (Fig. 1c, d, Fig. 2). Serpentine mineral stoichiometry from electron probe microanalysis (EPMA) indicates variable mixtures of lizardite with increasing proportions of talc accompanying increasing carbonation; samples 43z4 and 44z1 plot almost entirely between lizardite and brucite compositions while samples 44z2 and 44z3 extend towards talc to much higher Si and lower Fe + Mg (Fig. 3a). The mineralogy of the samples thus reflects carbonation via the intermediate reactions:

$$\underset{[lizardite]}{2Mg_3Si_2O_5(OH)_4} + 3CO_2 \rightarrow \underset{[magnesite]}{3MgCO_3} + \underset{[talc]}{Mg_3Si_4O_{10}(OH)_2} + 3H_2O$$

$$\tag{1}$$

$$\underset{[talc]}{Mg_3Si_4O_{10}(OH)_2} + 3CO_2 \rightarrow \underset{[magnesite]}{3MgCO_3} + \underset{[quartz]}{4SiO_2} + 3H_2O \tag{2}$$

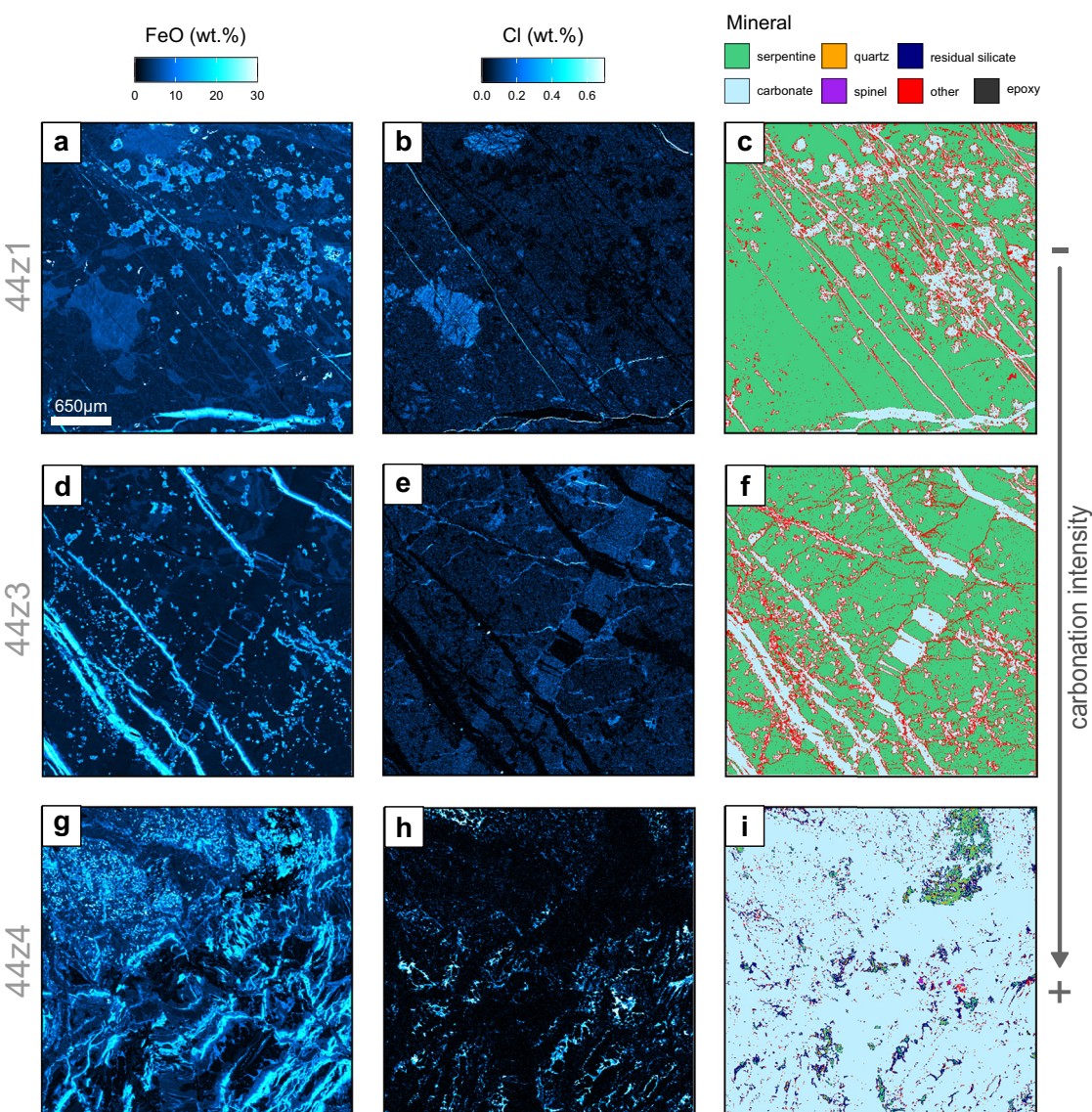

**Fig. 2 | Electron-probe maps of FeO (left) and Cl abundance (centre) and interpreted mineral maps (right) for three representative areas across the reaction zone. a–c** Sample 44z1 – least carbonated ophicarbonate showing serpentinite with small isolated patches of carbonate growth. **d–f** Sample 44z3 – more carbonated ophicarbonate, showing abundant patches of carbonate and development of parallel sets of Fe-rich magnesite veins. **g–i** Sample 44z4 – fully carbonated listvenite showing complete textural reorganisation relative to ophicarbonates, with zoned magnesite veins forming the majority of the sample; an amorphous, Cl-rich silicate phase fills the interstices between the carbonates.

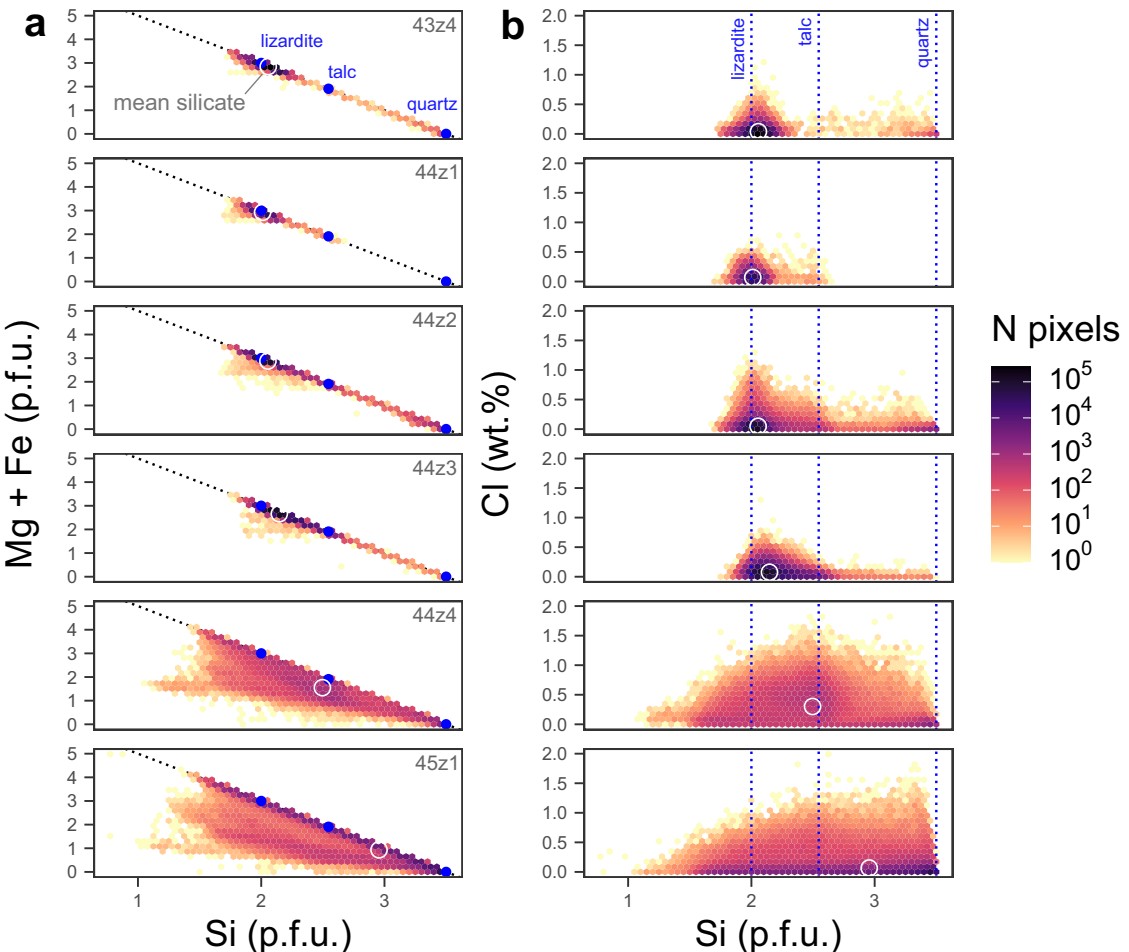

**Fig. 3 | Stoichiometry of serpentine and other silicate phases in reaction zone samples from quantified EPMA element maps with all silicate phase pixels plotted. a** Si atoms per formula unit (p.f.u; normalised to 7 oxygens) versus total Mg + Fe atoms p.f.u. showing increasing Si across the reaction zone progressing from lizardite to lizardite ± talc to increasingly non-stoichiometric talc-like compositions in interstitial silicates in 44z4 and 45z1. **b** Serpentine Cl versus Si p.f.u. showing a sharp drop off in maximum Cl contents away from stoichiometric lizardite (2 Si/7Ox.). Interstitial silicates in samples 44z4 and 45z1 show higher Cl contents for their Si content but make up a small proportion of the overall sample. The mean silicate composition for each sample is indicated by a white circle.

There is a systematic relationship between stoichiometry and Cl content in the ophicarbonates with the highest Cl concentrations closest to stoichiometric lizardite and lower Cl contents with increasing talc or brucite content, respectively (Fig. 3b). This suggests that Cl is principally hosted in lizardite and is less compatible in other minerals[30].

Carbonate is common in all four ophicarbonate samples and appears as scattered small (<10 μm) round growths of magnesite (Fig. 2a–c); larger patches (from 50 μm to several mm across) showing a rough globular or botryoidal habit; and as veins (Fig. 2d–i).

In the listvenites, magnesite is the principal constituent, with subordinate quartz, dolomite and, in samples from the lower listvenite (56z4, 60z3, 64z2), fuchsite. The listvenite samples from the reaction zone (44z4 and 45z1) both consist of a network of closely spaced subparallel veins. The EPMA mapping shows these typically have a symmetrical composite structure with a thin Fe-poor medial line of magnesite surrounded by an Fe-rich layer of either ferro-magnesite or Fe-oxyhydroxides, which typically grade into or are overgrown by Fe-poor magnesite at the edges of the veins (Fig. 4). The listvenite groundmass typically consists of small granular crystals of magnesite together with larger, often lozenge-shaped, dolomite crystals. Textural evidence from this and other studies indicates that dolomite is a late forming phase[19,31–33]; it primarily occurs either overgrowing/cross-cutting zoned magnesite-ferromagnesite veins or as an interstitial phase (Fig. 4). Interstices between the carbonates are either open voids or filled by a Cl-rich, poorly crystalline silicate phase whose composition departs from stoichiometric serpentine ± talc (Fig. 3b). Chlorine abundance is high in this phase (<2 wt%) but its overall abundance in the samples is low (<4 vol%), representing concentration of Cl into a minor residual phase as it is progressively excluded during carbonation.

**Chemical changes accompanying carbonation**

Bulk Cl, Br and I, show the highest concentrations in ophicarbonates, reflecting their relative compatibility in serpentine minerals ($Cl_{mean}$ = 264 ± 160 [1σ] ppm, $Br_{mean}$ = 428 ± 112 ppb, $I_{mean}$ = 32 ± 4 ppb, $n$ = 5; Supplementary Data 1). Listvenites show much lower abundances but higher Br/Cl and I/Cl ($Cl_{mean}$ = 22 ± 12 ppm, $Br_{mean}$ = 42 ± 13 ppb, $I_{mean}$ = 7 ± 5 ppb, $n$ = 7). Whole rock F concentrations show the opposite trend and are lowest in the reaction zone ophicarbonates ($F_{mean}$ = 28 ± 16 ppm, $n$ = 4) and slightly higher in listvenites ($F_{mean}$ = 35 ± 12 ppm, $n$ = 7). Metamorphic sole samples are highly varied, with relatively high F ($F_{mean}$ = 672 ± 583 ppm, $n$ = 4), low Cl and Br, but with high I/Cl ratios ($Cl_{mean}$ = 14 ± 4 ppm, $Br_{mean}$ = 21 ± 4 ppb, $I_{mean}$ = 51 ± 73 ppb, $n$ = 3).

In situ data from secondary ionisation mass spectrometry (SIMS) similarly show moderate to high halogen concentrations in serpentine ($F_{mean}$ = 172 ± 116 ppm, $Cl_{mean}$ = 614 ± 559 ppm, $n$ = 95; Supplementary

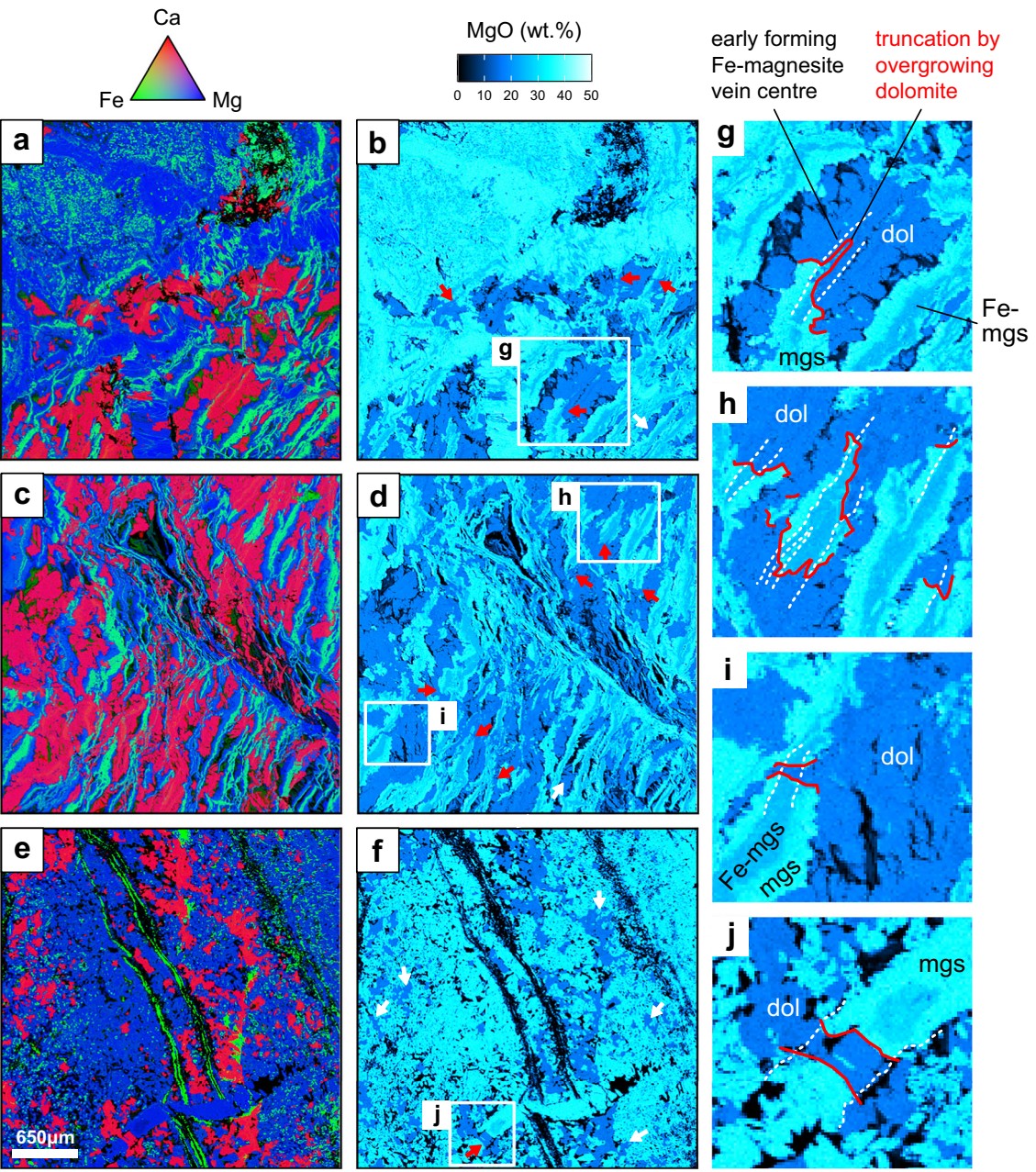

**Fig. 4 | Electron-probe element maps of representative areas in listvenite samples, showing evidence for a late stage dolomite formation.** Left: composite RGB image showing Fe, Mg and Ca abundance. Centre and right: MgO abundance for the same areas. **a**, **b** and **c**, **d** Sample 44z4 showing network of veins with Fe-rich magnesite cores and Fe-poor rims. Dolomite overgrows and crosscuts this texture. **e**, **f** Sample 45z1 showing dolomite forming interstitially between magnesite grains. **g**–**i** Annotated detail of four areas from panels **b**, **d** and **f**. Examples of dolomite crosscutting magnesite veins and filling interstices are indicated by red and white arrows, respectively, on MgO maps.

Data 4). Carbonates from both ophicarbonates and listvenites show generally slightly lower F and much lower Cl concentrations than serpentine ($F_{mean}$ = 117 ± 174 ppm, $Cl_{mean}$ = 121 ± 133 ppm; $n$ = 108). However, dolomite and some magnesite from the listvenites have markedly elevated F contents (dolomite $F_{mean}$ = 240 ± 172 ppm, $Cl_{mean}$ = 90.7 ± 249 ppm; $n$ = 20).

The bulk halogen contents of the ophicarbonates show strong negative correlations with bulk $CO_2$ showing expulsion of volatiles from the samples with progressive carbonation (Fig. 5). For Cl there is rapid initial loss followed by a more gradual decrease. We interpret this as a two-stage process, but the data can also be fit by an exponential curve ($Cl = a \cdot e^{-k CO2}$) which yields a similar or higher rate of early Cl loss (Supplementary Text 1, Supplementary Data 8). By contrast, bromine,

iodine and water show a relatively linear loss with $CO_2$ content, without any inflection in the slope. Chlorine is initially lost at much higher rates than Br or I, likely due to incompatibility of Cl in talc[30] (Fig. 3). This results in expulsion of a fluid with very low Br/Cl and I/Cl ($2 \times 10^{-4}$ and $1.6 \times 10^{-5}$, respectively; see Supplementary Text 1), during the early part of the carbonation reaction (Fig. 5). Addition of this expelled fluid thus changes the composition of the original carbonating fluid during the course of reaction.

## Fingerprinting the origin of carbonating fluids

Serpentine and carbonate from ophicarbonate samples are compositionally similar in F–Cl space (F/Cl = 0.1–1; Fig. 6). Carbonates from listvenites are also mostly colinear with serpentine compositions,

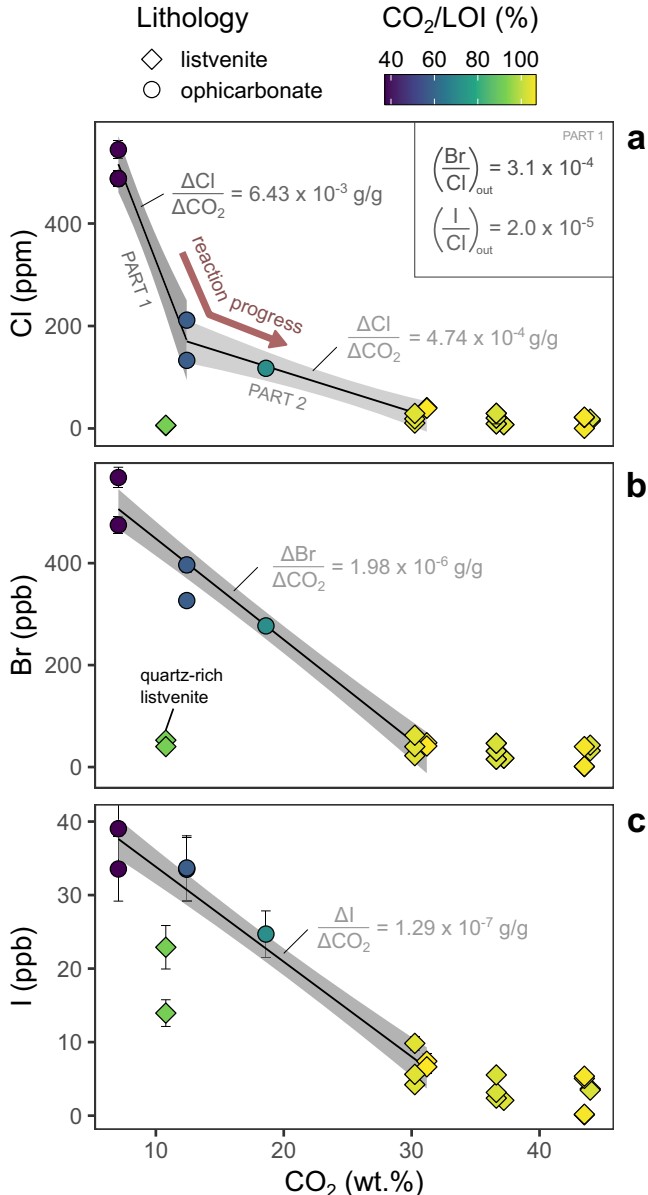

**Fig. 5 | Changes in bulk halogen abundance accompanying carbonation. a** Cl versus $CO_2$ showing rapid initial Cl loss interpreted as two distinct stages of reaction with different slopes. All data are colour coded by their $CO_2$/LOI ratio (i.e., the proportion of their total volatiles constituted by $CO_2$). **b** Br versus $CO_2$ showing a single consistent slope (**c**) I versus $CO_2$ similarly showing constant decrease with $CO_2$ increase. The Br/Cl and I/Cl of fluids expelled during carbonation of serpentinite can be calculated as the ratio of the respective Δhalogen/Δ$CO_2$ slopes (see Supplementary Text 1). These are shown for part 1 of the reaction as an inset to panel (**a**). Error bars (1σ) are mostly smaller than their respective data points.

although a subset show higher F concentrations. Serpentine halogen compositions show a progressive shift across the reaction zone mirroring visual, mineralogical and chemical indications of increasing equilibrium with $CO_2$-bearing fluids (Figs. 1–4). Bromine/Cl and I/Cl in individual spot analyses of serpentine in ophicarbonates form a correlated array ($R^2 = 0.52$) extending from very low Br/Cl and I/Cl (Br/Cl ≥ $2.50 \times 10^{-4}$; I/Cl ≥ $2.37 \times 10^{-5}$) over two orders of magnitude to higher ratios (Br/Cl ≤ $7.71 \times 10^{-3}$; I/Cl ≤ $1.35 \times 10^{-3}$; Fig. 7). The high Br/Cl and I/Cl end of this trend intersects the trend of sedimentary pore fluids[34]. The bulk compositions of both ophicarbonate and listvenite samples are also colinear with this array, with the mean listvenite composition plotting at the high Br/Cl and I/Cl end.

These combined observations indicate that both carbonate and serpentine compositions in the ophicarbonates are controlled by a common process and compositional endmembers. This suggests that the composition of serpentine across the reaction zone has been variably influenced by interaction with carbonating fluids and can therefore provide insight into their origin.

Serpentine Br/Cl and I/Cl are lowest in the samples furthest from the listvenite front (43z4 and 44z1) and increase towards the mean bulk composition of listvenites in the samples nearer the listvenite front (44z2, 44z3; Fig. 7). Both bulk Br/Cl and individual spots measured by SIMS show statistically significant correlations with bulk $CO_2$ ($R^2 = 0.81$, 0.16, respectively), the latter affected by intrasample scatter but nonetheless statistically significant ($p = 1.4 \times 10^{-7}$, $n = 165$). The low Br/Cl and I/Cl ratios observed in the ophicarbonates are below the seawater Br/Cl ratio ($3.47 \times 10^{-3}$) but above seawater I/Cl ($3.04 \times 10^{-6}$) and show distinctly lower Br/Cl than seafloor or forearc serpentinites[24,35] (Fig. 7b). We interpret the correlation between the two ratios as a binary mixing trend between a low I/Cl endmember, representing the uncarbonated serpentinite (or a fluid it has equilibrated with) and a high I/Cl endmember representing the composition of the carbonating fluid.

The main control on Br/Cl and I/Cl in serpentine is addition or loss of Cl; Br/I is nearly invariant across ophicarbonate samples to within analytical uncertainty (Br/I$_{bulk} = 13.29 \pm 2.35$, n = 5; Br/I$_{in-situ} = 9.64 \pm 5.11$) and this ratio therefore likely reflects the source of carbonating fluid. The average serpentine Br/I is consistent with sedimentary pore fluids which cover a wide range of Br/I with a mean of $3.0 \pm 9.8$[34]. In comparison, the underlying metamorphic sole shows lower Br/I of $1.77 \pm 1.42$ ($n = 3$) while altered oceanic crust shows a highly variable Br/I with a mean of $74 \pm 144$ ($n = 108$)[26,30,36,37], with most samples plotting to lower I/Cl than the ophicarbonate trend. The mean Br/I of the ophicarbonates is towards the higher end of the range of sedimentary pore fluids, suggesting fluids were derived from sediments with low I and therefore relatively low organic contents[38]. This is consistent with evidence from carbon isotopes in the listvenites which indicate an inorganic sedimentary source for the carbon[17].

However, the unusual low Br/Cl and low I/Cl serpentine endmember requires further explanation since it is significantly distinct from seafloor[35,39], forearc[35] and ophiolitic serpentinites[24], all of which have Br/Cl close to or higher than seawater (Fig. 7b).

## Carbonation-induced fractionation of halogens explains exotic serpentine compositions

It is possible that the unusual composition of serpentine in the least carbonated samples reflects an atypical serpentine chemistry characteristic of the Oman mantle section. However, serpentinites sampled far from the basal thrust are more typical of seafloor or forearc serpentinites, with Br/Cl close to seawater[40] (Fig. 7b). This strongly implies that the particular composition of the reaction zone ophicarbonates relates to carbonation.

Carbonation of serpentinite results in an expulsion of Cl-rich fluid (Fig. 5). As a result, fluids involved in the reaction should evolve to lower $CO_2$ contents, higher salinity and lower F/Cl, Br/Cl and I/Cl in a transient pulse preceding the carbonation front. Fluids fractionated in this manner therefore provide a potential source for the unusual halogen composition of the ophicarbonates, either by equilibration/recrystallisation of serpentine with the fractionated fluid or by direct serpentinisation. Notably, a similar trend of correlated Br/Cl and I/Cl, is shown by variably carbonated Isua serpentinites, with low Br/Cl compositions ($> 7 \times 10^{-4}$) suggested to arise from fractionation of fluids during water-rock reaction[41].

Mass balance would suggest the existence of a complementary high Br/Cl component. This may be represented by a handful of serpentine data and one bulk listvenite which have very high Br/Cl and/or I/Cl, suggesting such a component is volumetrically small and

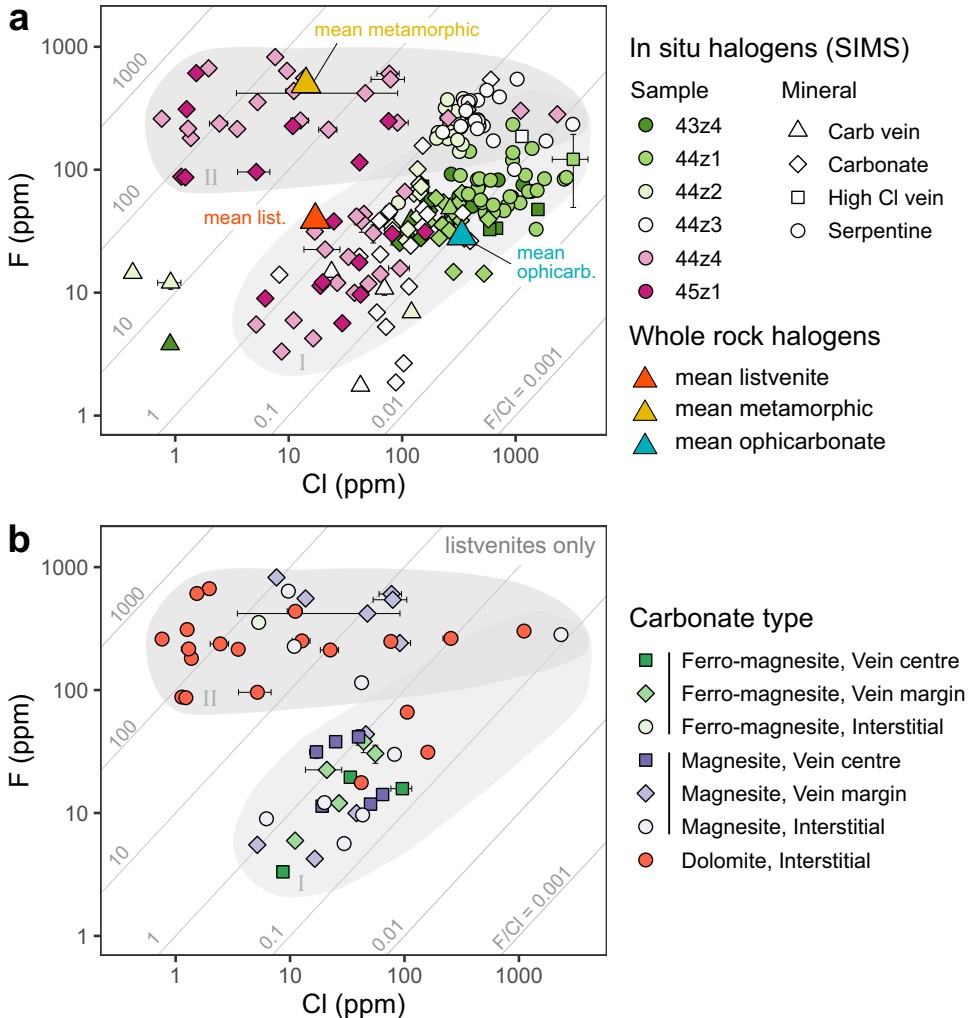

**Fig. 6 | In-situ halogen contents of carbonate and serpentinites. a** log-log plot of F versus Cl showing all data from reaction zone samples. The mean whole rock composition of listvenites, ophicarbonate and metamorphic sole samples is shown for comparison. The data fall into two arrays: one with relatively constant F/Cl (~0.01–1) and one with elevated F and variable F/Cl (1–1000), labelled I and II respectively. **b** As for panel (**a**) but only showing data from listvenites, colour coded by carbonate mineral and occurrence. Almost all early forming ferro-magnesite falls in the lower array while almost all late forming dolomite falls in the upper array. Error bars (1σ) are smaller than the symbols for most data points and, where visible, appear as asymmetric due to the logarithmic scale.

potentially poorly preserved. Such a component will only form at low fluid/rock ratios however (see below) and we therefore expect the average composition of the main body of listvenite to largely reflect that of the carbonating fluid (potentially with some contribution from the uncarbonated serpentinite protolith).

### Modelling fluid evolution to constrain its carbon and chlorine contents

Serpentinites are thought to faithfully reflect the relative halogen abundance of the fluids from which they form[35]. We therefore take the average composition of serpentine as reflecting the composition of a fractionated fluid. Since addition of carbon to the rock causes the release of Cl which modifies the chemistry of the fluid, fractionation at a given fluid/rock ratio will be more significant at lower fluid Cl concentrations and at higher $CO_2$ concentrations. Modelling the fluid fractionation required to produce the serpentinite signature therefore allows us to place compositional constraints on the carbon and Cl content of the fluid prior to carbonating reaction-driven fractionation. Since carbonate growing in reaction zone ophicarbonates is magnesite-dominated and serpentinite and most magnesite from all samples are colinear in F–Cl space (Fig. 6), the chemical

evolution of the ophicarbonate serpentinites is taken to be related to the main stage of magnesite dominated carbonation (rather than a later phase of dolomite-dominated carbonation, see below).

Modelling the evolution of carbonating fluid compositions can reproduce the data trend and halogen composition of the least carbonated samples' serpentine. However, we show that to do so requires the initial fluid to have high $CO_2$/Cl ratios.

If we define $R_{Cl}$ as:

$$R_{Cl} = \frac{\Delta[Cl]_{rock}}{\Delta[CO_2]_{rock}} \tag{3}$$

And, equivalently, define $R_{Br}$ and $R_I$, the Br/Cl of fluids expelled from the serpentinite during carbonation can be calculated as:

$$\left(\frac{Br}{Cl}\right)_{out} = \frac{R_{Br}}{R_{Cl}} \tag{4}$$

It can then be shown that the minimum $CO_2$/Cl ratio needed to explain a given final fluid Br/Cl ratio (Br/Cl$_f$) from its initial composition

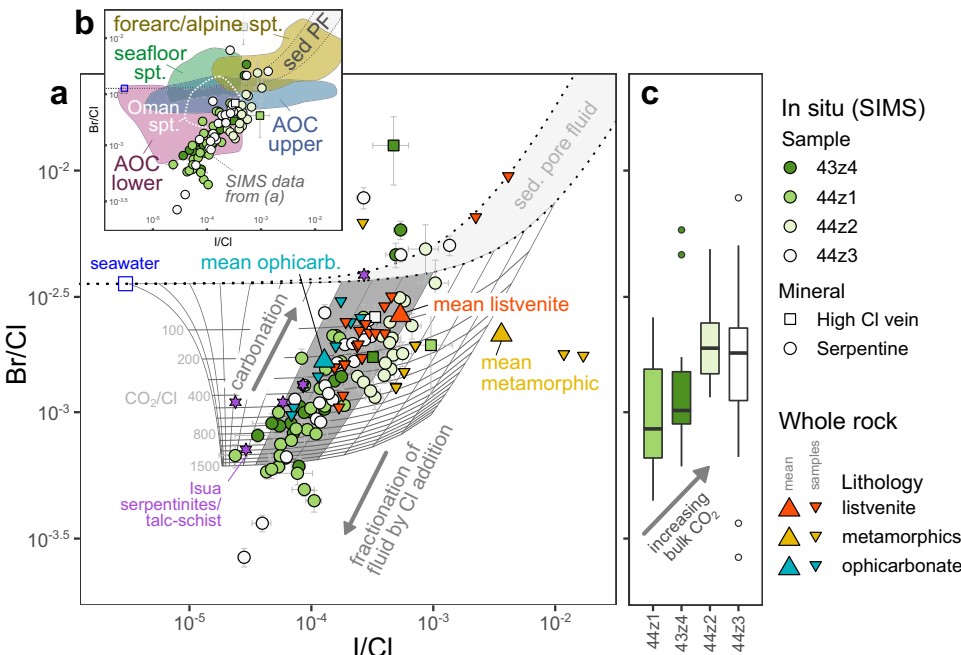

**Fig. 7 | In-situ heavy halogen composition of serpentine. a** Br/Cl versus I/Cl showing all in situ serpentine data, with 1σ error bars, compared with modelled fluid compositions. The composition of seawater and the range of sedimentary pore fluid are shown for comparison. Bulk data, including duplicate analyses, are shown for listvenites, ophicarbonates and metamorphic samples as inverted triangles and the mean composition for each lithology as large triangles. Data for variably carbonated Isua serpentinites[41] are shown as purple stars and the vector of carbonation for this study and Isua data indicated with an arrow. The results of modelling the change in fluid composition due to fractionation of the halogens form one another during carbonation is shown in grey for a range of starting compositions (taken as sedimentary pore fluid with variable initial I/Cl). The subset of solutions which provide a good fit to the trend of the data is highlighted as a dark grey area. **b** Inset showing the compositional fields of geochemical reservoirs which fail to account for the low Br/Cl serpentine compositions in BT1b. Shown are: forearc/alpine serpentinites[24,35], seafloor serpentinites[35], upper and lower AOC[26,30,36,37] and the Oman mantle serpentinites far from the basal thrust[40]. **c** Boxplot of SIMS serpentine data for each sample from (**a**) showing progressive increase in Br/Cl with increasing bulk CO2. The box indicates the interquartile range ($q_{0.25}$–$q_{0.75}$), and the thick horizontal line indicates the median. Outliers (> 1.5·IQR from nearest of $q_{0.25}$ or $q_{0.75}$) are shown as individual points.

(Br/Cl$_i$) is given by:

$$\frac{[CO_2]_{fluid}}{[Cl]_i} = \frac{1}{R_{Cl}} \left( \frac{\left[ \left(\frac{Br}{Cl}\right)_i - \left(\frac{Br}{Cl}\right)_{out} \right]}{\left[ \left(\frac{Br}{Cl}\right)_f - \left(\frac{Br}{Cl}\right)_{out} \right]} - 1 \right) \tag{5}$$

An equivalent expression can be derived by substituting I/Cl for Br/Cl (for full derivation see Supplementary Text 1). Since the least carbonated serpentinites have been affected by interaction with fractionated, Cl-rich fluid, this approach assumes that the compositional gradients across the reaction zone had reached a steady state (i.e., less carbonated samples evolve to resemble more carbonated samples; see Supplementary Text 1). If this is not the case it is possible that the Br/Cl$_{out}$ calculated is an underestimate and the initial CO2/Cl of the carbonating fluid required would be even higher.

The preceding calculation yields a minimum CO2/Cl since the exact fluid/rock ratio is not known; instead the minimum possible fluid/rock ratio required to explain the degree of carbonation observed is assumed (~1 to 23; Supplementary Data 8). For higher fluid/rock ratios the degree of fractionation in the final fluid would be less at constant CO2/Cl and, vice versa, a higher CO2/Cl ratio would be needed to explain a given degree of fractionation.

It is reasonable, however, to suppose that fluid rock ratios are likely to have been relatively low within the lens of remnant, partially carbonated serpentinites and ophicarbonates whose edge is transected by the reaction zone[18]; if these rocks had experienced high fluid/rock ratios they would have been carbonated like the rest of the mantle section at Site BT1b. Low fluid/rock ratios aren't necessarily representative of the entire succession. Rather, the reaction zone provides a fortuitous window into fluid fractionation at low fluid/rock ratios which allow us to constrain the fluid composition.

To calculate the CO2/Cl ratio of the fluid we take the average serpentine composition measured from sample 44z1 as representative of the most fractionated final fluid composition (Br/Cl = 1.1 × 10$^{-3}$, I/Cl = 1.0 × 10$^{-4}$). For an initial fluid composition close to that of sedimentary pore fluid with Br/Cl of 3.0 × 10$^{-3}$ and I/Cl of 3.1 × 10$^{-4}$, the model predicts very similar minimum CO2/Cl ratios of 394 and 407 to match the target Br/Cl and I/Cl, respectively during Part 1 of the reaction (Fig. 5; <12% CO2). Our calculation assumes a perfect closed batch reaction. However, Mg/Si ratios show evidence for mesoscale mobility of elements during carbonation[20]. Minor halogen redistribution may therefore explain the more extreme low Br/Cl and I/Cl ratios from individual spot measurements (Fig. 7).

These calculations indicate that carbonation was driven by fluids with markedly higher CO2/Cl than sedimentary pore fluid (400 vs. 0.02; assuming seawater salinity, 19,000 ppm[42] and Cretaceous CO2 contents, 360 ppm for the latter[43]) but with initially pore fluid-like halogen abundance ratios. This in turn implies an extra input of relatively carbon-rich and/or low salinity fluid. Mixing of pore fluid with a carbon-rich endmember (XCO2 of 0.01 to 0.2) can provide a fluid which fulfils the geochemical constraints (Supplementary Data 8).

The mixed carbonating fluid XCO2 is predicted to have been ~10$^{-1}$ to 10$^{-2}$, consistent with the upper end of estimates of the minimum XCO2 required to stabilise magnesite + quartz at 50 MPa and 100 °C, which range from ~10$^{-4}$ to 10$^{-2.5}$ [11,17,44]. The mineral assemblage of the listvenites does not provide a strong constraint on the XCO2 of the carbonating fluid provided it is above this minimum[11]. However, fluid-vapour immiscibility will occur at increasingly high CO2 contents, above XCO2 ~ 0.1 (at 275 °C and 0.1 GPa[45]). Our calculations meanwhile

suggest $XCO_2 \sim 0.05$ as an absolute minimum required to produce the fractionation seen. These two bounds therefore constrain the fluid $XCO_2$ to 0.05–0.10, notably similar to the fluid $CO_2$ content (20,000 ppm, equivalent to $XCO_2 \sim 0.01$) calculated by Kelemen et al.[21] from thermodynamic considerations.

The relatively low I contents of the carbonating fluid and lack of light C isotopic signatures rule out organic matter as the carbon-rich endmember[17,18]. Similarly, the maximum $CO_2$ content of low salinity meteoric waters ($CO_2/Cl < 6$) is too low to explain the degree of fractionation seen[46]. Suitable $CO_2$-rich fluids could, however, derive from carbonate dissolution or decarbonation in the subducting slab beneath the proto-ophiolite (Fig. 8).

Expelled slab fluids evolve to greater $XCO_2$ with depth in the subduction zone where shallow expulsion of pore fluid is followed by devolatilisation of water and then carbonate[47,48]. Carbon saturation in fluids in equilibrium with carbonate-bearing sediments at the approximate P-T conditions of listvenite formation has been estimated at 50–500 ppm[4]. However, at higher pressures the solubility of $CO_2$ increases such that fluids may contain up to 3 wt% $CO_2$ ($XCO_2 \sim 0.01$) at eclogite facies conditions[4]. Migration of such fluids into the shallow forearc could supply high $CO_2/Cl$ fluids. Similarly, decarbonation reactions are capable of supplying significantly more carbon-rich fluids ($XCO_2 \leq 0.2$)[49].

For a fluid resulting from dissolution of carbonate with 3 wt% $CO_2$ our calculation only yields a solution for very low Cl contents in the carbon rich endmember (Supplementary Data 8). Low salinity waters are found in forearc settings, derived from the breakdown of clay minerals in the slab[50]. However, breakdown of clay with 20 ppm Cl and 7% water[30] would be expected to generate fluids with ~300 ppm Cl, significantly in excess of those required by the model. Moreover, final fluid Cl is predicted to be only 35–75 ppm which would be expected to result in very low serpentine Cl contents (cf. mean $Cl_{serpentine} = 665$ ppm). Fluids from decarbonation reactions ($XCO_2 = 0.05–0.2$) yield model solutions with higher Cl contents (285–915 ppm) more consistent with those in serpentine. The latter appears to be the more consistent scenario with respect to the available geochemical constraints although we do not rule out high pressure carbonate dissolution.

Whether the involvement of a carbon-rich fluid from metamorphic decarbonation necessitates significant lateral movement of fluids depends on the depth of both carbonation and decarbonation reactions, both of which are uncertain. The PT conditions of listvenite formation are poorly constrained due to a lack of pressure-sensitive mineral parageneses, but can be broadly constrained to 0.3 to 1.4 GPa (~10–50 km)[18,27,51]. Pressure estimates based on clumped isotope temperatures[17,19] rely on assumptions of geothermal gradient but suggest similar maximum depths of 10–50 km with lower temperatures potentially representing a protracted cooling history[19,21].

Modelling of devolatilisation reactions in the downgoing slab generally indicate that the main metamorphic release of both $CO_2$ and $H_2O$ occurs in a pulse between 2.2 and 2.6 GPa, corresponding to depths of ~60–80 km in the forearc[49,52,53]. This would imply lateral flow of fluids into the forearc. However, the depth of decarbonation will also depend on the thermal structure of the subduction zone, with warmer subduction favouring decarbonation, commencing as shallow as 40 km[53–55]. The Semail subduction zone appears, at least initially, to have been hot as indicated by sediment-derived granitoid melts generated at granulite facies conditions[56]. This may have favoured metamorphic decarbonation at shallower levels. Overall the mixing of sedimentary pore fluid with carbon-rich fluids to produce the carbonating fluid implies lateral migration of fluids, since most pore fluid expulsion occurs relatively shallowly during subduction[47,48]. That said, the exact distance between these loci of devolatilisation and carbonation is difficult to constrain and may have been relatively small under a warm subduction geotherm.

However, despite this geochemical evidence requiring input of decarbonation-derived $CO_2$-rich fluids, a subduction origin for carbonating fluids is contradictory to interpretations from recent geochronology which has dated dolomite veins in Oman listvenites to 60 Ma and younger[22], significantly after the cessation of subduction at ~80 Ma[57,58]. Some resolution to this dichotomy is therefore needed.

## Resolving the dichotomy between geochemistry and geochronology

While most serpentine and carbonate analyses plot in a single array of consistent F/Cl, a subset of analyses show distinctly higher F contents (< 1000 ppm) which are decoupled from Cl contents (Fig. 6). Samples 43z4 and 44z1 fall entirely within the lower F/Cl array; however, the remaining samples (44z2, 44z3, 44z4, 45z1) straddle the two, with analyses in both groups.

Subdividing the carbonate data by mineral (e.g., dolomite, magnesite) and texture (Fig. 6b), reveals that nearly all dolomite analyses plot in the higher F/Cl array whereas all analyses from the early-forming ferromagnesite vein cores are restricted to the lower F/Cl array. Combined with the sample petrography, these observations provide robust temporal constraints on fluid composition; dolomite clearly overgrows the zoned ferromagnesite vein network and otherwise is typically an interstitial phase (Fig. 4). This indicates that an early low F/Cl fluid was responsible for the main phase of magnesite-dominated carbonation (Stage 1) and interacted with all the ophicarbonate samples to some degree, while a second, higher F/Cl fluid resulted in the growth of dolomite (Stage 2), cross-cutting earlier carbonate growth in the listvenites and affecting serpentine compositions only in ophicarbonate samples 44z2 and 44z3 (Fig. 6). The average composition of the listvenites lies close to the Stage 1 array but slightly offset towards higher F contents (Fig. 6), reflecting the small proportion of Stage 2 fluids which have contributed to their composition. The strong correlation of Br/Cl and I/Cl for all analyses (Fig. 7) suggests that the two fluid sources had similar Br/I ratios although there is a weak correlation between I/Cl and F/Cl which suggests the Stage 2 fluids were slightly more I-rich compared to Stage 1. Average Ca contents of listvenites and Oman mantle[20] recalculated and normalised as Ca, Fe or Mg carbonate give 8.7 and 1.3 wt% $CaCO_3$, respectively, suggesting that Ca metasomatism accounts for <10 wt% of the total carbonate. Otherwise, the similarity of mean listvenite Mg:Si:Ca ratios to those of uncarbonated Oman peridotites indicates that carbonation was largely isochemical for the major elements[17,20].

The occurrence of a second stage of carbonation with a different halogen signature and clear petrographic evidence indicating it postdates the main stage of magnesite-dominated carbonation is notable and provides a potential resolution to the dichotomy between geochemistry and geochronology. The dating of Scharf et al.[22] relies primarily on analyses of dolomite. Our observation that dolomite is a late-forming phase is corroborated by numerous other studies of listvenites from BT1b and nearby localities[19,31,33]. The identification of BT1b dolomites as belonging to a distinct carbonation event with a markedly different fluid composition therefore suggests that the young dates from other sites may represent minimum ages for the main phase (Stage 1) of carbonation, albeit with some inherent uncertainty in extrapolating results between localities. This would suggest that the bulk of carbonation occurred before 60 Ma, coincident with subduction, as indicated by a range of geochemical proxies – not least the sedimentary pore fluid halogen signature (this study) and carbon isotopes[18] of $CO_2$ bearing fluids.

It should also be noted that the bulk chemical characteristics of the BT1b listvenites suggest that they are highly unsuitable for U-Pb dating. All the listvenites and associated serpentinites have very low U/Pb ratios and lie at the extreme end of a trend defined by forearc serpentinite compositions reflecting low fluid U contents under low $fO_2$ conditions[20,59]. A priori, it would be expected that any attempt to date such materials by U-Pb methods would be inherently biased towards later perturbations under more oxic conditions where U would be more

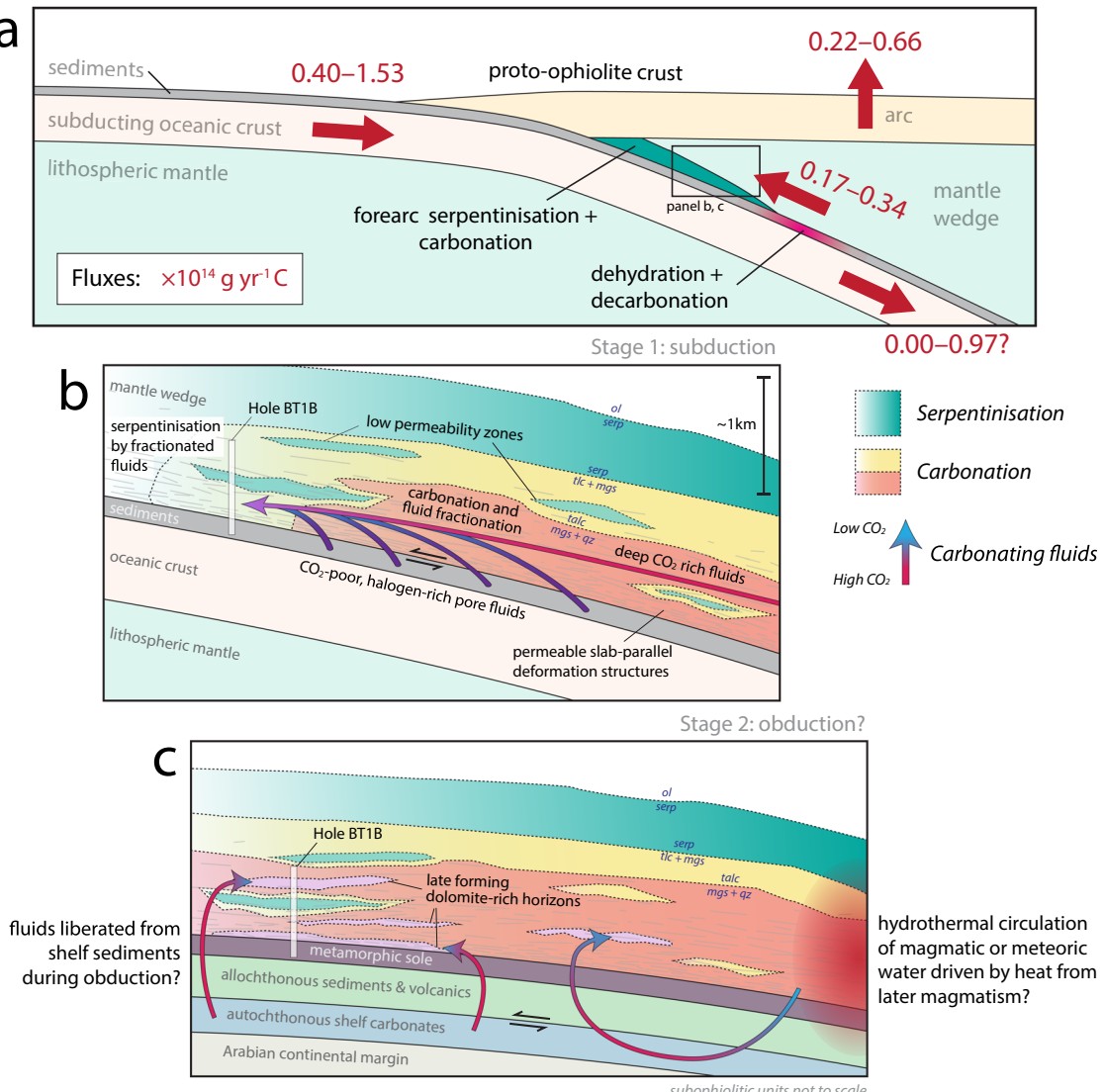

**Fig. 8 | Schematic diagram illustrating the tectonic setting, estimated carbon fluxes, and interpreted sequence of carbonation. a** Schematic of the inferred tectonic setting for Stage 1 carbonation. Large red arrows indicate the major $CO_2$ fluxes (in $10^{14}$ g yr$^{-1}$ C): (from left to right) entering subduction[4,61,62], return flow into the forearc (this study), volcanic and diffuse arc outfluxes[4,62], net flux to the deep mantle (this study). **b** Stage 1 carbonation – fluids, derived from a mixture of deep decarbonation reactions and sedimentary pore fluid expulsion, carbonate the forearc mantle. Downstream, fractionated fluids depleted of $CO_2$ cause serpentinisation, setting up moving fronts of carbonation and serpentinisation. Some low porosity lenses escape reaction. **c** Stage 2 carbonation – F and Ca rich fluids, possibly from either obduction-related metamorphism or later magmatic activity, form dolomite and magnesite of a different halogen composition along discrete horizons. Sub-ophiolitic units after Searle et al.[28].

soluble in fluids. We tentatively suggest that, unless their listvenite bulk geochemistry differs significantly from that at BT1b, a sequence of later fluid infiltration events may be what was dated by Scharf et al.[22].

Rather than a single event we envisage formation of the listvenites as a multi-stage process: prior to carbonation the mantle peridotite protolith appears to have been partially serpentinised possible in a seafloor setting[40] (Fig. 7b). Subsequently the BT1b mantle section was incorporated into the forearc of a young subduction zone (Fig. 8a). Water- and carbon-rich fluids derived, respectively, from sediment compaction and decarbonation/dissolution reactions deeper in the subduction zone reacted with the serpentinites, forming talc and eventually carbonate. This released Cl from serpentine, lowering the Br/Cl and I/Cl ratios of the fluids. These fractionated fluids resulted in additional serpentinisation (or re-equilibration with existing serpentinites) further along their flowpath, producing unusually low Br/Cl and I/Cl serpentine (e.g., 43z4, 44z1; Fig. 7). Where Stage 1 carbonation reached completion, removal of aqueous carbon and release of Cl

ceased and the carbon content and Br/Cl and I/Cl of the fluids 'downstream' of them increased, such that moving fronts of carbonation and serpentinisation were established (Fig. 8). Later, Stage 2 dolomite and magnesite crystallisation occurred from a Ca-rich fluid with higher F/Cl and I/Cl ratios. This stage is only dominant in some horizons and microstructurally crosscuts earlier carbonates. Fluids derived from low temperature metamorphism of autochthonous shelf carbonates during obduction[51] are a plausible source for the F- and I-rich Stage 2 fluids, especially given the marked enrichment of Ca in the lowermost listvenites overlying the basal thrust[20]. Their similarity in halogen composition to the metamorphic sole may relate to interaction with allochthonous units, the protoliths for the metamorphic sole[20,29]. Equally, Stage 2 carbonates could be derived from interaction with another F-rich reservoir (e.g., fluid circulation associated with later igneous activity[23,60]). Implicitly this suggests that Stage 2 dolomite-dominated carbonation likely occurred at equal or lower pressure (~0.3 GPa[51]) than much of Stage 1, although this is difficult to confirm.

Overall these results are unsurprising if ophiolitic mantle sections are viewed as a patchwork of overprinting metasomatic events (e.g., ref. 24) not all of which may be apparent or detectable at all scales or at all localities. It is likely that multiple stages of fluid infiltration affected the ophiolite prior to, during and since obduction and may have overprinted some chemical or isotopic systems. Furthermore, the extent of such overprinting may have varied spatially resulting in different conclusions from studies at different sites (e.g., ref. 18 cf. 22). Nevertheless, our data indicate a strong subduction affinity which has escaped such overprinting and attest to fluxing and carbonation of the shallow forearc by $CO_2$-rich slab derived fluids.

## Implications for subducted carbon fluxes

Having demonstrated strong evidence for a subduction origin for the bulk of carbonation expressed at Site BT1b we may consider the implications for subducted $CO_2$ fluxes (Fig. 8a). Listvenites are widespread although not continuously exposed throughout the ophiolite (Fig. 1a). The metamorphic sole is also not ubiquitously preserved and in numerous places late brittle thrusting has juxtaposed relatively unmetamorphosed allochthonous sediments against the mantle section[29]. This suggests that listvenites might also be prone to removal during obduction and may therefore have relatively poor preservation potential, especially in less laterally extensive ophiolites. In the broader geological record there are numerous examples of listvenites associated with the basal thrusts of supra-subduction zone (SSZ) ophiolites although often with some ambiguity as to the timing of their formation relative to obduction[11]. Nonetheless they are not ubiquitous in SSZ ophiolites with exposed metamorphic soles, which suggests either that they are poorly preserved or that specific conditions are required for their genesis.

Modelling of fluid evolution (see above) indicates that pore-fluid provides 48–87% of the Cl in the carbonating fluid (Supplementary Data 8). This allows us to constrain the deep $CO_2$ flux through mass balance and independent estimates of pore-fluid fluxes in subduction zones. Much of the initial pore fluid contained in subducting sediment is lost at very shallow depths and expelled through the accretionary wedge. By 10 km depth the instantaneous flux of pore fluid from sediment is of the order of $1 \times 10^{13}$ g/yr globally[61]. Mass balance of the composition of the carbonating fluid, assuming the proportion of Cl from pore fluid (48–87%), a pore fluid chlorinity of 0.11–0.15 wt% and pore-fluid flux of $1 \times 10^{13}$ kg/yr into the forearc[61], yields a deep Cl flux of $4.5 \times 10^{10} - 1.6 \times 10^{11}$ g/yr. Combined with calculated $CO_2$/Cl ratios of $CO_2$-rich fluids (Supplementary Data 8) this corresponds to a deeply sourced carbon flux into the forearc of $1.7 \times 10^{13} - 3.4 \times 10^{13}$ g C/yr. It is possible that expelled pore-fluids will not be perfectly mixed with deeper-sourced fluids so this likely represent an upper bound. Nonetheless the magnitude of the potential flux is a significant proportion (11–86%) of estimates of the total subducted carbon flux, which range $4.0 \times 10^{13} - 1.5 \times 10^{14}$ g/yr [4,61,62]. Combined with estimates of diffuse and volcanic arc outfluxes[4,62] this suggests limited deep subduction of $CO_2$, unless subduction inputs are towards the upper end of their estimated range. If carbonation is ubiquitous in the shallow forearc at subduction zones, as has been suggested on the basis of gases in forearc springs[8], it would constitute a significant proportion of the global input $CO_2$ flux which is neither subducted to the mantle, nor returned to the atmosphere.

It should be noted, however, that listvenites have not been directly observed in dredges or drilling at modern forearcs such as Mariana and Tonga which allow fluid compositions at slab depths to be inferred. This highlights key differences between those and the Semail subduction zone which may promote listvenites genesis in the latter, including: subduction geotherm (cold[63] vs. hot[27]), sedimentary input (dominantly siliceous, 2.6% $CO_2$[64] vs. mixed volcanic-carbonate-siliciclastic, 8.6% $CO_2$[18,65]), and even redox state and the relative timing of serpentinisation and carbonation[66] (see Supplementary Text 2). This raises the important possibility that the magnitude of the forearc carbon sink may vary spatially and temporally in response to changes in oceanic sedimentation, subduction zone thermal regimes and redox state over geological time and could therefore play an important role in fluctuations in the long-term carbon cycle.

# Methods

## Sample preparation

For the samples investigated here, two immediately adjacent billets were sawn from each sample and one polished as a thick (-130 μm) section for in situ analyses (EPMA, SIMS, see below) and one powdered for bulk rock halogen and mineralogical analyses. Powders were produced from -10 g of rock, which was first ultrasonicated in distilled water, coarsely crushed in a porcelain pestle and mortar and then powdered by hand in an agate pestle and mortar to <125 μm particle size. Thin sections were set in epoxy resin, then polished using water.

## Bulk halogens by pyrohydrolysis

A split of whole rock powders was analysed for bulk halogens (F, Cl, Br, I) at Institut de Physique du Globe de Paris, (IPGP), Paris, France. Full details of method and analytical procedures are given in ref. 67. A -500 mg aliquot of powdered sample was mixed with the same amount of $V_2O_5$ and placed in a platinum crucible. The sample was directly introduced into a quartz combustion tube and heated at 1200 °C in a $H_2O$-vapour stream driven by a nitrogen gas flux. The $H_2O$-vapour containing halogen acids extracted by hydrolysis was condensed via a cooling system and trapped into a vial containing 10 mL NaOH solution (25 mMol L$^{-1}$). Vapour flux was adjusted in order to complete extraction in -45 min to 1 hr with a final solution volume -80–100 mL. The mean dilution factor (mass of solution/mass of sample) was -150 to 200. F and Cl were determined by ion chromatography and Br and I by ICP-MS. Both methods were conducted with standard calibration curves bracketing the range of halogen contents in sample and rock standard solutions. For IC these were prepared by dilution of commercial standard in deionised water with 2% v/v $HNO_3$. For ICP-MS the calibration solutions were likewise prepared from commercial standards and matrix matched to the final solutions from pyrohydrolysis (5% v/v NaOH, 2% v/v $HNO_3$) to avoid any potential matrix effects[67]. In both cases calibrations were linear for all elements analysed. Repeat analyses of rock standards (BHVO-2, GS-N, JG-1, JR-1, JB-1b, JB-2) indicate mean relative errors of -10% for F and Cl, -30% for Br and -60% for I, and yields of >95% for all halogens[67]. Detection limits in solution were well below sample concentrations for F and Cl (< 100 μg/L) and were -10 ppb and 1 ppb for Br and I, respectively. For extraction of 500 mg of sample, producing 100 ml of pyrohydrolysis solution, detection limits in rock samples are estimated to be 20 mg kg$^{-1}$ for F and Cl and 2 μg kg$^{-1}$ for Br and I. The mean analytical errors on the studied composition ranges were estimated at 10 mg kg$^{-1}$ for F and Cl, 100 μg kg$^{-1}$ for Br and 25 μg kg$^{-1}$ for I. No blank correction was necessary as procedural blanks (both tube and tube + platinum crucible) were always below detection limits. The batch of $V_2O_5$ used was tested prior to analysis to ensure it did not contain detectible halogens. Repeat analyses were conducted on three samples (44z1, 56z4, 60z3) to assess repeatability. Relative standard deviation (RSD) between the repeats ranged from 3–10% for F, Cl and $SO_4^{2-}$. RSD for Br and I varied, with one sample (44z1) showing 7% variation, one showing Br and I below detection (56z4) and one showing 70–80% RSD suggesting either intra-sample heterogeneity or less reliable analytical reproduction of Br and I. A summary of all whole rock halogen data is given in Supplementary Data 1 and the pyrohydrolysis dataset is provided in Supplementary Data 2.

## Bulk halogens by the noble gas method

Heavy halogens (Cl, Br, I) were also measured using the noble gas method[68,69]. Prior to irradiation, -20 mg of finely crushed (< 250 μm) sample was wrapped in aluminium foil and sealed in an evacuated

quartz tube. Standards were spaced throughout the tube to monitor neutron conversion of K, Cl, Br and I. Samples were irradiated for 48 hours in the flooded reflector area of the Missouri University Research Reactor Centre (MURR), Missouri, USA from 23$^{rd}$–24$^{th}$ March 2022, at a predicted flux of $8 \times 10^{13}$ n cm$^{-2}$s$^{-1}$. Combined data from monitors spaced along the irradiated tube were used to account for minor vertical variations in neutron flux. A Lagrange polynomial fit to Hb3Gr monitor data was used to interpolate noble gas–element conversion factors for each sample based on their vertical positions in the irradiation tube.

Halogens were determined from their proxy noble gas isotope abundances ($^{38}$Ar$_{Cl}$, $^{80,82}$Kr$_{Br}$, $^{128}$Xe$_I$) measured using an ARGUS VI multi-collector noble gas mass spectrometer at the University of Manchester. Prior to analysis, 1–5 mg of each sample was loaded into an aluminium sample holder baked at ~120 °C for at least 12 hours to achieve high vacuum. Samples were heated to fusion in a single step under vacuum using a Photon Machines CO$_2$ fusion laser (Model C-55L Laser HD), with a 3 mm diameter defocused beam. Where possible, two repeat analyses were made for each sample. Laser powder was gradually ramped from 0.7–13.2 W over 180 s to achieve fusion without overly vigorous degassing of samples. A subset of 3 samples was analysed by crushing *in vacuo*; ~20 mg of sample was loaded into stainless steel manual valve-type crusher, baked as above and the gas released by crushing in 1 or 2 steps. The crushed residues were retained and analysed by laser heating as above. Released sample gases were purified for 10 minutes by exposure to SAES NP10 and ST172 getters at room temperature and ~300 °C, respectively, and the purified noble gases were then expanded into the mass spectrometer. During the analytical session blanks were measured at the start of the day and then between every 1–2 analyses. A calibrated aliquot of air was measured daily to monitor instrument performance, mass discrimination and sensitivity. Due to the elapsed period between irradiation and analysis, $^{37}$Ar$_{Ca}$ had entirely decayed away and a small correction for the decay of $^{37}$Ar to $^{36}$Ar was instead made based on the Ca content measured by XRF. Details of data processing steps were otherwise all as in refs. 24 and 68. A summary of key halogen and noble gas data is given in Supplementary Data 1 and the full NI-NG-MS halogen and noble gas dataset including all data processing steps is provided in Supplementary Data 3.

**Secondary ionisation mass-spectrometry (SIMS)**

In situ concentrations of halogens (F, Cl, Br, I) and carbon were determined by secondary ionisation mass spectrometry (SIMS) using a Cameca IMS 1270 (NERC Ion Microprobe Facility, University of Edinburgh, UK). Sections were re-polished with a 1 μm diamond paste to remove carbon coating from EPMA analyses, cut into rounds, ultrasonicated in distilled water, dried and finally sputter-coated with a ~30 nm layer of gold.

The SIMS analyses were carried out over two sessions, first for F and Cl and then for Br and I. Fluorine and Cl were determined in serpentine, carbonates and a range of veins from all six samples (totalling 208 spot analyses). A Cl-bearing interstitial silicate phase in samples 44z4 and 45z1 could not be analysed due to its poor polish giving surface irregularity and electric field gradients that can lead to increased mobility of halogens and the risk of electrical arcing within the instrument.

Bromine and I could only be determined in serpentine and high Cl veins from samples 43z4, 44z1, 44z2, 44z3 (92 spots total) due to low concentrations in carbonate and the lack of a reliable peak centring species (a result of the variable carbonate chemistry). Bromine and I were determined adjacent to the F-Cl pit using petrographic images and EPMA maps to ensure significant heterogeneities were avoided and the second pit was as much as possible within the same grain/mineral domain as the first. A subset of points in samples 44z1 and 44z2 were analysed at a third adjacent point to determine carbon together with repeat F and Cl measurements (59 spots total).

The ion microprobe was operated with a 10 kV Cs$^+$ primary ion beam with a negative secondary accelerating voltage of −10 kV resulting in a nominal beam size of 1 μm at ~1 nA. Fluorine and Cl were determined simultaneously with a 60 s pre-sputter, rastered over a $15 \times 15$ μm square, to remove gold coat and any surface contaminants and 10 cycles of analysis measuring $^{18}$O, $^{19}$F, $^{30}$Si, $^{35}$Cl, on an electron multiplier with count times of 2 s, 4 s, 2 s and 4 s, and settling times of 1 s, 1 s, 9 s and 2 s, respectively. Backgrounds were measured at masses 17.5 and 17.8 during the magnet settling time, with a total time (including settling) of 15 s. A 2 mm field aperture was used. Analysis for C, F and Cl was identical except for the addition of $^{12}$C with 2 s counting and 1 s wait times, and background/settling steps at masses 11.2 and 11.5 for 6.5 s each.

The $^{19}$F and $^{35}$Cl signals were internally normalised to both $^{18}$O and $^{30}$Si but $^{18}$O gave better stability throughout the analytical session and better agreement with Cl contents determined independently by EPMA. Fluorine and Cl concentrations were calibrated against laboratory glass standard T1G (F 321 ppm, Cl 113 ppm, SiO$_2$ 58.60 wt%) using the $^{19}$F and $^{35}$Cl relative ion yields (RIY) to $^{18}$O determined on 5 repeat analyses each morning. It should be noted that some serpentine analyses had Cl significantly above that of the primary standard. If signal response was not linear with increasing Cl this could result in large systematic errors. However, linearity of the signal was assessed by analysis of two natural glasses with different SiO$_2$ contents and Cl contents (Supplementary Figs. 1, 2; T1G, 58.6 wt% SiO$_2$, ATHO, 75.6 wt% SiO$_2$). Processing data from each standard using the other as the primary standard gives F and Cl abundances that are within 10% of the accepted value for each. The standard deviation on the calculated relative ion yields was propagated together with the analytical error on each point and includes the increase in uncertainty involved with extrapolating beyond the calibration range. Moreover, the SIMS Cl data can be checked against EPMA data from the same points and show a strong linear corelation with a slope of 1.04 (Supplementary Fig. 3). The combined evidence therefore points to the reliability of these data. Nonetheless, the potential for unaccounted-for systematic error in the highest Cl analyses should be noted.

Bromine and I were analysed under similar conditions as above, but at a mass resolution of ~22,000 and with a 1.5 mm field aperture. The surface was pre-sputtered for 30 s and 10 cycles of $^{30}$SiO$_3$, $^{79}$Br, $^{81}$Br and $^{127}$I measured on an electron multiplier with count times of 2 s, 5 s, 5 s, and 5 s and wait times of 2.5, 1.5, 2.5, and 1 s, respectively. Backgrounds were measured at masses 76 and 126 for 60 s and 40 s, respectively, to allow the magnet to stabilise after large mass jumps. Analyses were internally normalised to $^{30}$SiO$_3$. Reported F/Cl, Br/Cl and I/Cl ratios use the $^{18}$O-normalised Cl values as discussed above. The linearity of the Br signal over a range of concentrations was established by repeat analyses of a range of halogen-bearing standards (Supplementary Fig. 4; scapolites, BB1, BB2, SY; Kendrick et al.[69]; USGS synthetic glasses GSD, GSE; Myers et al.[70]; Marks et al.[71]). Some analyses of samples showed significantly higher $^{81}$Br/$^{79}$Br ratios than their natural isotope ratio (0.977) suggesting an isobaric interfering species on $^{81}$Br in some analyses. No samples showed unusually low $^{81}$Br/$^{79}$Br, implying the measurements of $^{79}$Br are unaffected by such an interference. As a result, Br contents have been calculated from $^{79}$Br only and scaled to 100% from its natural abundance (50.57%). As I is monoisotopic, similar assessment of possible interferences could not be made. Bromine and I concentrations were calibrated to the composition of USGS synthetic glass GSE (267 ppm Br, 25.06 ppm I, 53.7 wt% SiO$_2$[71,72]), which provided the closest SiO$_2$ matrix match to our serpentine samples in a reference material with known I and Br abundances. Relative ion yields of Br and I were determined from three repeat analyses at the start of each analytical session and their standard deviation propagated with the analytical uncertainty. For serpentine, median relative uncertainties on measured abundances were 2.5%, 2.4%, 7.4% and 17% for F, Cl, Br and I and on ratios were 4.7%, 8.4% and 17.3% for F/Cl, Br/Cl and I/Cl,

respectively. The uncertainties on carbonate analyses were slightly higher with median values of 4.2%, 5.4% and 6.9% for F, Cl and F/Cl respectively. For both carbonate and serpentine, all analyses had relative uncertainties below 15%, 23%, 26% and 25% for F, Cl, Br and I respectively and below 26%, 26% and 35% for F/Cl, Br/Cl and I/Cl, respectively. The largest analytical errors were from analyses of thin high-Cl veins, likely due to these typically sampling some surrounding material. Median uncertainties were 59%, 35%, 31% and 36% for F, Cl, Br and I, respectively, and 69%, 48% and 50% for F/Cl, Br/Cl and I/Cl.

Matrix effects are a potential issue in SIMS analysis[73]. These effects are difficult to assess or correct for since there are no widely available matrix-matched standards available for halogens in serpentine or magnesite. Nonetheless, the good agreement of SIMS, EPMA and NI-NG-MS data (see below, Supplementary Fig. 3) lends confidence that any matrix effects are relatively minor as do the linear responses of F, Cl and Br across a range of standard matrices in this study (Supplementary Figs. 1, 2, 4) and others in the literature[74]. Furthermore, differences in matrix effects are likely to be minimised, although not absent, in analyses of the same mineral, meaning that relative differences in compositions of a given mineral can be compared with some confidence. The ratio of F and Cl relative ion yields for two natural glasses with a wide range of $SiO_2$ contents are within 20% of one another suggesting matrix effects cannot explain the large variations seen in data (carbonate F/Cl = 0.02 to 398, serpentine F/Cl = 0.02 to 1.48). Furthermore, within carbonates, the principal control on F/Cl appears to be textural rather than mineralogical with interstitial dolomite and magnesite both showing high F/Cl (Fig. 6b). Nonetheless, it is impossible to discount that some matrix effects may affect the data and particularly absolute concentrations. As a result, care should be taken in making direct comparison of concentrations between different minerals.

A summary of SIMS halogen data is provided in Supplementary Data 4 and the full dataset is given in Supplementary Data 5. SIMS standard data are provided in Supplementary Data 9.

## Comparison of halogen datasets

Comparison of bulk halogen data collected by NI-NG-MS and pyrohydrolysis reveals some notable differences. Chlorine abundances generally agree reasonably well however, the abundance of Br and I is lower in most pyrohydrolysis analyses suggesting low yields during pyrohydrolytical extraction, possibly related to the unusual sample matrix. Monitoring sample-by-sample yield requires the use of hazardous radioisotope tracers (e.g., $^{125}I$)[75] and was not performed in this instance, so this cannot be confirmed.

It is difficult to directly compare SIMS and NI-NG-MS data since these were measured on individual minerals and bulk rocks chips, respectively. Nonetheless, the Br/Cl and I/Cl ratios are all within error of the mean of SIMS data from serpentine (which is likely to dominate the whole rock signature). Moreover, bulk NI-NG-MS data and individual serpentine SIMS data are colinear in Br/Cl–I/Cl space suggesting any slight differences arise due to differences in the proportions of different serpentine endmember compositions analysed by SIMS versus the bulk composition of the sample. The SIMS chlorine data also agree well with results from EPMA (Supplementary Fig. 3). The correspondence of halogen data from NI-NG-MS, SIMS and EPMA shows the consistency of the data across multiple methods.

A significant proportion of listvenite samples had Br and I below detection by pyrohydrolysis. This makes calculation of a mean composition problematic and prone to systematic error depending on the assumptions made about the distribution of the data (e.g., log-normal, gaussian, uniform). In producing an overall bulk halogen dataset, the NI-NG-MS data are preferred for the heavy halogens (Cl, Br, I) since they provide a complete dataset for all samples analysed. These data are supplemented by the F abundance from pyrohydrolysis. To avoid any potential inaccuracy due to analytical differences between the two methods, the F/Cl ratio for each sample was calculated using the Cl abundance measured by pyrohydrolysis.

## Analysis of major elements, total carbon and loss on ignition

Major elements were determined by X-Ray Fluorescence Spectroscopy (XRF) on fused glass beads at the Earth Surface Research Laboratory, Trinity College Dublin. Beads were prepared by mixing powdered sample and lithium borate flux (50:50 LiT:LiM) in a Pt crucible with a sample:flux ratio of 1:10, before fusion and casting using an automated Claisse LeNeo system at 1050 °C.

Major element data were acquired on a Malvern Panalytical Zetium wavelength dispersive XRF spectrometer using a calibration based on WROXI synthetic standards and optimised using international certified reference materials. For full methodological details and accuracy and precision of analytical results determined by repeat analysis of secondary standards see Carter et al.[76].

Total carbon (TC) was determined by combustion on a separate sample aliquot using an Analytik Jena multi EA4000 element analyser at the Earth Surface Research Laboratory, Trinity College Dublin. A primary calibration curve was prepared using a blank boat and 6 aliquots of differing mass of NIST SRM 1632b (Trace elements in coal) and was optimised for the expected range of TC in samples (-0.1–12 wt%). Pure $CaCO_3$ (Analytik Jena, 12 wt% TC) was used as a secondary standard to monitor instrument performance; repeat analyses indicate relative accuracy of better than 3% and relative precision of better than 5%. Depending on expected sample TC abundance, 250–400 mg of sample powder was weighed into a clean ceramic boat, and the weights of both sample and total of sample + boat recorded. TC was determined by combustion in $O_2$ at 1200 °C for ~10 minutes and measurement of $CO_2$ on a nondispersive infra-red (NDIR) detector. Following TC analysis, loss on ignition (LOI) was determined from the combusted sample + boat by gravimetry. The full major element, LOI and TC dataset is provided in Supplementary Data 6.

## X-Ray diffraction (XRD)

A split of each of the whole rock powders used for bulk halogen analysis was analysed for its mineral constituents by XRD at the University of Manchester. Powders were mixed with ~1 ml of amyl acetate and spectra were acquired using a Bruker D8 Advance Diffractometer. CuKα1 X-rays were produced from a copper target with a wavelength of 1.5406 Å and the diffracted spectrum scanned from 5–70° 2θ, with a step size of 0.02° and a count time of 0.2 s per step. The resulting spectra were matched to a database of reference mineral spectra using the EVA software package (version 4). The XRD spectra were then further processed to semi-quantitatively calculate the relative proportions of minerals in the samples. The full dataset mineral abundance is provided as Supplementary Data 7.

## Electron probe microanalysis (EPMA)

In the six reaction zone samples (43z4, 44z1, 44z2, 44z3, 44z4, 45z1), target areas for in-situ analysis by SIMS were quantitatively mapped by EPMA using a Cameca SX100 Electron Microprobe at the University of Manchester. The operating conditions were 15 kV accelerating voltage and 200 nA beam current with a 5 μm electron beam. Maps of $SiO_2$, $Al_2O_3$, FeO, MgO, CaO, Cl, $K_2O$, $Na_2O$ and $Cr_2O_3$ were produced for 24 areas of $2.56 \times 2.56$ mm with a step (i.e., pixel) size of 5 μm. Fluorine was not mapped due to its high detection limits and the detrimental impact counting times would have had on the analysis of the other elements. Maps were fully matrix-corrected and quantified as both silicate normative (0.5 calculated $O^{2-}$ per $X^+$ cation) and carbonate normative (0.5 calculated $CO_3^{2-}$ per $X^+$ cation) using the ProbeFor-EPMA software package.

All pixels (each corresponding to a single point analysis) were extracted from the maps using a script written in the R programming language and used to assess mineral chemistry and stoichiometry

across the reaction zone. Pixels corresponding to serpentine or carbonate phases were extracted by filtering by the following compositional criteria (all in wt%, applied in order from top to bottom):

- $SiO_2 < 5$ & $Cl > 0.25$: *epoxy*
- Total $< 70$ & $SiO_2 < 15$: *carbonate*
- Total $> 70$ & Total $< 92$ & $CaO < 1$ & $SiO_2 > 35$: *serpentine*
- Total $> 90$ & $Cr_2O_3 > 0.5$ & $SiO_2 < 5$: *spinel*
- $SiO_2 > 85$ & $CaO + MgO + FeO < 25$: *quartz*
- (in listvenite samples) $SiO_2 > 20$: *residual silicate*
- Everything else: *other*

where other mostly consists of non-ideal mixed analyses along the boundaries between minerals.

Structural formulae, normalised to 7 oxygens, have been calculated for each pixel to interrogate serpentine stoichiometry. Given the EPMA beam diameter of 5 μm, intergrown microcrystalline minerals are not spatially resolvable and the analysis will represent a mixture of the individual constituents. Plotting mineral compositions as Si/7 oxygens against Fe+Mg/7 oxygens allows discrimination between lisardite, brucite, talc, quartz and chlorite and microcrystalline mixtures of these endmember components[77].

Compositions for individual SIMS points (for internal normalisation to O or Si and validation of Cl data) were extracted by averaging the pixels (~60) in a polygon drawn round each point location in QGIS software. Where spots were analysed in areas not covered by an EPMA map, an average composition of petrographically similar areas was used for internal normalisation. Within each sample there was only minor variation between points with typical relative standard deviations of 4% on O and $SiO_2$ in serpentine and 6% on O in carbonate.

## Data availability
All data generated and analysed in this study, including raw and summarised geochemical data and modelling, are included in Supplementary Data 1–9.

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

## Acknowledgements

The authors would like to thank John Craven and the Edinburgh Ion Microprobe Facility for setting up and assisting with in situ halogen analysis, and Marguerite Godard and Thierry Decrausaz for helpful and illuminating discussions. This work was supported by the Natural Environment Research Council through a PhD studentship to E.C. [NE/L002469/1] and through access to the Edinburgh Ion Microprobe Facility [IMF663/0518]. B.O'D acknowledges current research support from the Natural Sciences and Engineering Research Council of Canada (NSERC Discovery Grant) and from the Newmont Chair in Economic Geology (University of Ottawa). This research used samples and data provided by the Oman Drilling Project. The Oman Drilling Project (OmanDP) has been possible through co-funding from the International Continental Scientific Drilling Project (ICDP; Kelemen, Matter, Teagle Lead PIs), the Sloan Foundation–Deep Carbon Observatory (Grant 2014-3-01, Kelemen PI), the National Science Foundation (NSF-EAR-1516300, Kelemen leadPI), NASA–Astrobiology Institute (NNA15BB02 A, Templeton PI), the German Research Foundation (DFG: KO 1723/21-1, Koepke PI), the Japanese Society for the Promotion of Science (JSPS no:16H06347, Michibayashi PI; and KAKENHI 16H02742, Takazawa PI), the European Research Council (Adv: no.669972; Jamveit PI), the Swiss National Science Foundation (SNF:20FI21_163,073, Früh-Green PI), JAMSTEC, the TAMU-JR Science Operator, and contributions from the Sultanate of Oman Ministry of Regional Municipalities and Water Resources, the Oman Public Authority of Mining, Sultan Qaboos University, CNRS-Univ. Montpellier, Columbia University of New York, and the University of Southampton.

## Author contributions

E.C. designed the study with assistance from B.O'D., E.C. selected and prepared all samples with assistance from P.C. and R.B., analysis was conducted by E.C., R.B., H.B.B. and P.B., E.C. took the lead in interpreting results and wrote the original draft, with B.O'D., R.B., P.C., and H.B.B. assisting with editing and comments, the Oman Drilling Project Science Team acquired drill core, logged the core, and collected contextualising geochemical and mineralogical datasets.

## Competing interests

The authors declare no competing interests.

## Additional information

## the Oman Drilling Project Science Team

**Elliot J. Carter** ⓘ [1,2] ✉

For a full list of members and their affiliations see ref. [16].

