## [Transparent Peer Review file · Nature Communications]

Carbonated mantle peridotites represent a hidden sink for subducted CO₂

Corresponding Author: Dr Elliot Carter

Version 0:

Reviewer comments:

Reviewer #1

(Remarks to the Author)

The authors present a detailed mineralogic, petrologic, and geochemical (focused on C and halogens) of variably carbonated serpentinites from Oman (through a drill core to the metamorphic sole). The goal is to determine fluids involved in carbonation reaction, carbonation of the forearc, and the subsequent role of the forearc in the C cycle.

The authors have collected an impressive amount of data linking mineralogy (and different mineral generations) to geochemistry (particularly detailed in situ work). This is no small feat! In fact, determining the different chemistry of the dolomite is key to interpreting the data in context with previously determined geochronology (I thought this was a particularly nicely done part of the contribution that demonstrated the importance of detailed contextual work).

Major comments:

I recognize that halogen work is extremely difficult, and particularly halogen concentrations via SIMS is limited. Therefore, some more details on methods (see comments below) would be helpful. Also given that some of the same data are collected by different methods, and in some cases have dramatically different results, these differences must be addressed and why some data are used and not others.

The discussion linking dehydration, carbonation, and depths of these reactions needs to be clarified and better linked to the proposed (2 step) model from the outset.

Nature Communications is targeted for a general geology audience. There are numerous places throughout (see comments below) in which a few more details would help the contribution be more accessible. In particular, including other reservoirs (e.g., serpentinites, AOC, mantle) on figures (Fig 7) would be helpful in following the interpretation/discussion.

Minor comments:

Given that Nature Communications allows for 70 references and the authors only have 54 references, many more references can be included particularly in the introductory section. For example, on lines 31-32, the authors state "Recently attention has turned to the potential for hidden outfluxes not sampled by volcanism and unaccounted for in current estimates^{6,7}."

Numerous additional references can be added to this, in which previous authors note commonly overlooked outfluxes via the forearc or backarc, as well as sequestration in volatiles in additional reservoirs (e.g., sub-continental lithospheric mantle). Some suggestions are:

Bekaert DV, Turner SJ, Broadley MW, Barnes JD, Halldórsson SA, Labidi J, Wade J, Walowski KJ, Barry PH. Subduction-driven volatile recycling: A global mass balance. *Annual Review of Earth and Planetary Sciences*. 2021 May 30;49(1):37-70.

Gibson, S. A., & McKenzie, D. (2023). On the role of Earth's lithospheric mantle in global volatile cycles. *Earth and Planetary Science Letters*, 602, 117946.

Barnes, J. D., Manning, C. E., Scambelluri, M., & Selverstone, J. (2018). The behavior of halogens during subduction-zone processes. The role of halogens in terrestrial and extraterrestrial geochemical processes: Surface, crust, and mantle, 545-590.

Foley, S. F., & Fischer, T. P. (2017). An essential role for continental rifts and lithosphere in the deep carbon cycle. *Nature Geoscience*, 10(12), 897-902.

Paragraph starting on line 48. At the start of the sentence clarify that you are discussing the listvenites from the Semail

Ophiolite. Is this discussing the exact same samples that you studied, or just from the same area?

Line 55: Geochronology based on what? More details are needed since this is related to one of the key motivations of the work.

Fig 1. Some of the small text in Fig 1a, 1d, and 1e is really hard to read. One could use mineral abbreviations in 1d and 1e to save space, thus allowing for a larger font size.

Fig 1. Show the location of all the samples in the figure, not just those in the reaction zone. Otherwise, it looks like there are only 6 samples.

Fig. 4. d-f and g-i are not labelled on the figure. It looks like it is the same sample in a row, but the caption reads like it is the same sample in a column?

Line 201: Are these EMPA or SIMS data or an average of both?

Line 231: Higher F contents than what? It looks like the serpentinite from the ophicarbonates has just as high F concentrations as the listvenites. Or do you mean that some listvenites have higher F concentrations than other listvenites?

Line 238: Since you have the space, you may want to add more references for sedimentary pore fluids. For example: Muramatsu, Y., Doi, T., Tomaru, H., Fehn, U., Takeuchi, R., and Matsumoto, R. (2007). Halogen concentrations in pore waters and sediments of the Nankai Trough, Japan: Implications for the origin of gas hydrates. *Applied Geochemistry*, 22(3), 534–556.

Line 246: Showing some statistics to back up this statement would be helpful (e.g., box and whisker plots for the different samples) because I don't see a trend. Overall, the data look highly scattered. For example, 43z4 has some of the highest and lowest Br/Cl values and some of the highest and lowest I/Cl values.

Line 250: "seawater serpentinites"- reword. I assume you mean abyssal serpentinites hydrated by seawater? Put serpentinites (both abyssal and forearc) on Fig 7 for reference. "Plot away from" is vague. Be more specific (and adding serpentinites to the figure will help).

Line 252: That would mean the serpentinite protolith has a very, very low Br/Cl (lower than the reported mantle value of $2.8 \pm 0.8 \times 10^{-3}$; Kendrick et al., 2017). How is this reconciled?

Line 258: Clarify the arrays. Do you mean samples with high F/Cl and with low F/Cl?

Line 264: A plot of Br vs I (showing different reservoirs) would be helpful.

Line 270: Add AOC onto Figure 7. AOC span a large range of I/Cl, many with high I/Cl values.

Fig 7. Given that the focus of the paper is on carbonation, it might be useful to add carbonate to this figure. Since iodate can substitute for carbonate, why is the I/Cl not higher (like as seen in some AOC)?

Fig 7. Why are there more whole rock data shown than samples? For example, there are 4 metamorphics listed on Table S1, but more than 4 metamorphic samples shown on Fig 7. Is this because Fig 7 is showing both Pyr and NI-NG-MS data? (If so, then there needs to be some discussion on why the different methods determine different concentrations- see below.)

Line 297: The full reference for reference 31 is not given. What are the halogen values for other Oman serpentinites (and their tectonic relationship to your samples)? Can you add these data to your figures for comparison?

Line 304: Since this is key to the presented interpretation, I suggest drawing this trend/arrow on the figure.

Line 308: Can you expand on this since it is directly related to your interpretation? Adding those data onto figure 7 (with an arrow depicting the trend) would be helpful

Line 335: And what is that fluid/rock ratio that is assumed?

Line 369: Reference?

Line 375: Clarify both scenarios? At shallower and deeper depths? What CO₂ concentrations are used at each of those depths? The assumed parameters are not completely clear. Are these different scenarios related to parts 1 and 2 on figure 5?

Lines 399-440: The volatile release is also going to be a function of the thermal structure of the subduction zone. What is most reasonable for your study area?

Line 403: Based on the discussion starting on lines 368, it is not clear exactly what the depths of carbonation are. This discussion linking dehydration, carbonation, and depths of these reactions should be streamlined and better linked to the

steps outlined in the model later in the manuscript.

Lines 404-405: This sentence seems like an add on.

Line 418: delete "etc" and add "e.g.," before dolomite

Line 462: Careful with the wording here because the serpentinites in this study do not have halogen ratios similar to those from other forearc regions (other regions have higher Br/Cl and I/Cl plotting up in the pore fluid range). (The subsequent explanation on the following lines on the evolving composition is much clearer here than previously in the manuscript. Some of these concepts need to be better introduced earlier to avoid confusion.)

Line 478: Reference? This is interesting, but if going to be mentioned then should be introduced earlier in the discussion.

Line 508: Define Prop CIPF

Line 550: A few more details on the bulk rock pyrohydrolysis + IC/ICP-MS halogen concentrations are needed. For example, what are the yields (or concentrations) of halogens on rock standards (e.g., Kendrick et al, 2018)? What are the blanks (e.g., halogen blanks from NaOH, V₂O₅, etc.)? Are blanks corrected for or are they all below detection limits? What were the analytical protocols for the IC and ICP-MS- or at least a reference to where these are outlined?

Line 584: Were rock standards analyzed via NI-NG-MS? Comments on the reproducibility of samples analyzed via NI-NG-MS? Given the poor reproducibility of Br and I on some samples via pyrohydrolysis and ICP-MS and the explanation was "sample heterogeneity or less reliable analytical reproduction of Br and I" it would be good to see if potential sample heterogeneity is also documented via NI-NG-MS.

A comparison of Br and I concentration data collected by the two different methods shows stark differences. The Br concentrations via pyrohydrolysis are approximately 50% less than those of the noble gas method, except for one sample with ~470% more Br. The same is seen with I, in which I by pyrohydrolysis records much lower concentrations except for a couple with ~400-500% more I. Some mention of the poor comparison is needed.

In Table S2 it says that Cl, Br, and I values determined by NI-NG-MS are preferred. Why? But, then in Table S1, it says that F and F/Cl uses F and Cl data measured by Pyr-IC. Why is the Cl measured by Pyr-IC used if NI-NG-MS is preferred? And is the Cl Pyr-IC data only used for the F/Cl and not [Cl]? In Table S2, it is denoted that Cl is measured via ICP-MS (not IC).

Line 617: Was only one glass standard used for Cl and F SIMS analyses? Is there concern about matrix effects given that serpentine, carbonate, and vein material was analyzed? On Table S5, both EMPA and SIMS Cl data are given. With the exception of some very high Cl concentrations via EMPA, there is reasonable agreement. This would be good to point out.

Table S1: What does "shallowwater-corrected data used for I abundance and I/Cl ratios" mean?

Reviewer #2

(Remarks to the Author)

This manuscript focuses on the carbon cycle in subduction zones, exploring the role of carbonated mantle peridotites in sequestering subducted CO₂ through a study of the Oman ophiolite. The research is of great significance, aiming to address the long - standing debate regarding the origin of carbonating fluids and their impact on the global carbon cycle. The approach combines multiple analytical techniques, which is innovative and promising. In general, I think this manuscript is well written. I don't have special comments for the authors. I would suggest acceptance for this manuscript.

Reviewer #3

(Remarks to the Author)

Carter et al. present a well written and illustrated paper integrating careful petrography with multiple geochemical techniques, several of which are innovative in their application to the topic. The paper presents a new, untapped line of evidence (halogens) and a compelling petrographic case for a two stage listvenite formation in the SE Oman ophiolite. This conclusion does the best job so far in the literature of reconciling the controversial mismatch between previous geochronological and geochemical interpretations. The authors take this conclusion and point out its implications for a hidden carbon sink in forearc mantle, which would represent an important component of subduction systems and the deep C cycle.

These are the now the most studied listvenites in the world and will doubtless continue to receive attention from the subduction zone cycling community if a forearc wedge carbonation interpretation is partly true. The rocks and reactions are also highly relevant to recent efforts to engineer peridotite carbonation as a means to permanently sequester CO₂. Halogens as important fingerprints, ligands, and catalysts in such reactions are an important but overlooked part of this effort.

The subject matter is of vibrant interest and the conclusions are of relevance to several different swaths of our community: those working on listvenites and ophiolite, those on subduction zone cycling, and those on global biogeochemical cycling.

I have a few concerns below and many minor comments, that should hopefully be possible to address with minor/moderate revisions.

General comments

The main argument I see against the principle thrust of the paper – that forearcs are a hidden carbon sink, is that listvenites are discontinuous along the metamorphic sole in Oman, and rarely present in other ophiolites with exposed soles (?), and have not been dredged from forearc sections in the Marianas or Tonga trenches. Can this be considered, explained or conclusions tempered to reflect this reality? Showing the locations of all the listvenites described in the ophiolite in a more detailed Fig 1a would assist this discussion.

Its important to note that Scharf et al. worked on listvenites from another locality, and that this study focuses only on listvenites from BT1B. While the drilled localituy is the among the largest listvenite bodies in the ophiolite, it is not the only one. If there were indeed two pulses of carbonating fluids, then it would follow that they have different spatial foci/extents, which might explain differing conclusions from workers in different parts of the ophiolite.

The conclusions of this paper rely heavily on the petrographic evidence for dolomite overprinting magnesite. I don't find the evidence presented very clear/convincing. Can it be expanded on or shown in higher resolution/different imaging types?

The final panel of Fig 8c and the accompanying metamorphic sole F-rich fluid source interpretation doesn't make sense. By the time of this F rich dol precipitation (~60 Ma by this model), the sole had already been metamorphosed to amphibolite facies (and exhumed). Exhumation and retrogression of these rocks cannot produce a fluid at shallow depths – they are almost dry following their trip into the mantle. The source must be tectonically deeper than the sole in lower grade rocks. There are also several nappes beneath the ophiolite that are omitted here, as 'Arabian continent margin' evokes the autochthonous continental basement. I recommend consulting the cross sections in Searle (2007, Geoarabia) to get the tectonostratigraphy correct, and this may help with your interpretation of F rich fluid source options.

Figure comments:

Fig. 1. The intervening met. sole close up section (Fig 1c) breaks the flow of the figure from left to right. Is it really necessary considering it has been well described and no samples in this study come from it? If you deem it so, it may be better to either side of the full 0–300 m stratigraphy.

Fig. 2. Annotate the side of the fig. with an arrow indicating carbonation intensity.

Fig. 4. I don't find the dol x-cutting particularly convincing, at least at the resolution in the merged PDF. Dol doesn't have euohedral faces, it looks like the Mgs is corroding early dol. The red vs white arrows don't seem to have differing significance?

How closely does the drill hole depth approximate the true pseudostratigraphic depth? This should be mentioned in the caption or annotation.

Fig. 8. IN panel a, it looks like the mantle wedge is dehydrating and decarbonating.

Line by line comments:

L45. The tectonic setting of the ophiolite does still attract controversy, a reference here would serve to illustrate which of the various models the authors prefer to contextualize this work.

L50. Adding specifics of the Sr ratios in parentheses would be helpful here. Also, for transparency, note that they overlap the Hawasina metasediments, but also other crustal materials present in the sub-ophiolitic sequence, i.e. they are also likely intermediate between the protolith and the more radiogenic autochthonous basement.

L55. There is certainly a mismatch in the interpretations of the trace elements and geochronology, the key point to made/tone to be struck is that the uncertainty/apparent nature of the dichotomy is not a fault of the data, but interpretation of them on one or both sides of the argument. Set out the timing constraints specifically here, as it is crucial context.

L63. State which halogens.

L78. What other volatile cycles does it have a bearing on?

L94. The origin of those metasediments/metabasalts is of importance for context here, as is the intervening deeply subducted and exhumed met. sole. See Searle's papers for reference.

L96. Reference for the Oman Drilling Project?

L97-101. *More tectonostratigraphically complete. This sequence is not a stratigraphy. Tectonostratigraphy/pseudostratigraphy are sometimes used in this situation. Or otherwise avoid the term altogether.

L107. How is the minimal impact of protolith heterogeneity established for these rocks? Sharp compositional variations between dunite/harzburgite are common near the sole.

L110. Reaction zone at the base of the upper listvenite? Or where?

L133. Along the reaction zone? This needs clarification (further into the reaction zone? With increasing carbonation?)

L132. Don't all three serpentine minerals have similar stoichiometry?

L217-219. Would it? This is not a closed system process, the reaction proceeds because fresh CO₂ rich fluids are introduced and spent ones (necessarily) removed, maintaining high CO₂ activity and high fluid/rock ratios. The implication would be Cl addition to serpentinites downstream. Is this seen?

L296. It is not surprising the Oman serpentinites have markedly different halogen compositions to seafloor serpentinites. Can a more relevant comparison be made to other ophiolitic serpentinites that would rule out the distinct possibility this is some common feature of deeply formed, ophiolitic serpentinites (or those formed by meteoric fluids?)

L306. What is meant by equilibration of a serpentine protolith? Recrystallization?

L407. It is more than 'somewhat' in conflict, at least for a single event carbonation model, which is worth specifying. Also, it should be ordered 100–80 Ma; however, recent petrochronological work from e.g. Garber et al. has failed to reproduce the >100 Ma Guilmette et al prograde dates, so it may be safer to say from 'at least 96 Ma to ~80 Ma' or to focus on the cessation date, which is the relevant one here.

L441. I agree this is significant, but the petrographic evidence as provided is not that clear/definitive. In line with my previous comment can this be bolstered?

L446. These samples were from a different locality, so I would not say that they have been identified (with certainty implied) as belonging to a distinct carbonation event. Your data rather provides a compelling explanation as to why they may belong to a distinct carbonation event and potentially represent minimum ages.

L451. It should be noted that this statement is limited in scope to BT1B, clearly some aspect of the listvenites (primary or secondary) was suitable to dating to Scharf et al.

L460. Is this parallel list not reversed?

L479. It would be a pertinent place to note a fact all too often forgotten by studies of these cores, that these rocks were transposed under a ~15 km thick ophiolitic nappe over 100s of km of continental margin (Searle et al. 2007) before reaching their final resting place. Rocks so close to the basal thrust could hardly escape progressive overprinting by metamorphic fluids during this translation.

L482: the ophiolite does not have 100 Myr subaerial exposure history, indeed, it is only 96 Ma and a large part of its life was underwater.

L484. Last line could be more specifically geared toward carbonating fluids. There is nothing controversial about forearc slab fluid fluxing.

Line 533: Spell out IPGP.

Version 1:

Reviewer comments:

Reviewer #3

(Remarks to the Author)

The Authors have done an exceptionally thorough job of addressing the comments, leaving tighter and more balanced/considered manuscript than previously. I congratulate them and recommend it for immediate publication.

Reviewer #4

(Remarks to the Author)

Carter et al. have conducted a detailed study of the mineralogy and halogen composition of ophicarbonates and listvenites from a drill core through the Oman ophiolite. The use of variation in halogen compositions in these samples to place constraints on fluid sources and the quantity of CO₂ introduced to the forearc mantle is quite novel. Both the results and the methodology used will be of interest to the subduction zone community as well as most geochemists studying volatile exchange between Earth's mantle and surface. This manuscript is well-reasoned and will be an excellent contribution. The authors addressed previous comments from reviewers adequately. I have a few additional concerns regarding a few assumptions made by this study and the methods used.

Comments on Main Manuscript:

Lines 222-223:

The negative correlation between bulk halogen content and bulk CO₂ content is clearly demonstrated by Figure 5, but the two-stage process for Cl loss is an interpretation of the data. This could also be a single-stage process with a non-linear correlation between Cl and CO₂. The wording here needs to be clear that this is an interpretation.

Lines 257-259 and Figure 7c:

I don't find this claim very convincing. Serpentine Br/Cl and I/Cl from all ophiocarbonate samples seem to span most of the range of Br/Cl and I/Cl. The boxplot in Figure 7c tries to show the trend with bulk CO₂, but I don't see much of a trend in the data. However, a statistical test of correlation or differences in the Br/Cl in different samples would be the best way to show this.

Lines 343-355:

To use the equation on line 353 to determine the CO₂/Cl of the fluid, this paper makes assumptions about the Br/Cl of the starting fluid, the final fluid composition, and how Cl and Br change with changing CO₂ content during Stage 1 of the model. In the model presented, an initial fluid with a sedimentary pore fluid-like composition alters the peridotite forming serpentine, which then reacts with CO₂ to form carbonates, preferentially expelling Cl over Br and resulting in the expulsion of a low-Br/Cl fluid. This fluid then travels through the peridotite farther from the CO₂ reaction front, altering it and imparting the low Br/Cl signature of the fluid onto those rocks. For the Br/Cl composition of the initial fluid in the equation on line 353, a sedimentary pore fluid-like composition is used because the high-Br/Cl-I/Cl end of the sample array in Figure 7 has a composition similar to sedimentary pore fluids, correct? But wouldn't mass balance require that expulsion of a fractionated, low-Br/Cl fluid result in the partially-carbonated partially-dehydrated residue having higher Br/Cl than the initial fluid that first formed the serpentine? So, the highest Br/Cl values observed in the samples would actually be higher than the composition of the initial fluid. Of course, exactly how fractionated the Br/Cl of the residual partially carbonated samples are would depend on the extent of halogen loss, but it could be quite extensive in the most CO₂-rich samples. Unless I am misunderstanding the assumption made about the initial fluid Br/Cl composition, then this pitfall needs to be addressed. Similarly, the calculation of (Br/Cl)_{out} essentially determines the Br/Cl of the fluid expelled from the rock during carbonation by determining the change in bulk rock Cl and Br contents for a given change in bulk CO₂ ($\Delta\text{Cl}/\Delta\text{CO}_2$; $\Delta\text{Br}/\Delta\text{CO}_2$) and then taking the ratio of these two ratios, correct? However, the inherent assumption made by this argument and in Figure 5 is that the ophiocarbonates with the lowest CO₂ and highest Cl, Br, and I contents have bulk Cl, Br, and I contents representative of the halogen contents originally in the higher CO₂ ophiocarbonates and listvenites prior to their more extensive carbonation and halogen loss. However, it is later argued that the ophiocarbonates with the lowest CO₂ contents have interacted with a fractionated fluid that has higher Cl and lower Br than the fluid that initially serpentinized rocks closer to the CO₂ reaction front. In that case, the Cl contents of the samples farther from the listvenites would be higher than the initial Cl contents of the samples closer to the listvenites, whereas the opposite would be true for Br. This would result in the RCl on line 347 being overestimated and RBr being underestimated, which would make the calculated value of (Br/Cl)_{out} lower than it should be. Again, perhaps I am misunderstanding an aspect of the assumptions made, but if I am not mistaken then the potential impact of this should be addressed.

Figure 6:

There are error bars with no associated data in both a) and b).

Comments on Methods Section:

Lines 604-605:

There are many potential problems with measuring halogens by pyrohydrolysis-IC-ICP-MS that need to be clear to the reader. As an example, if the calibration standard solutions are not appropriately matrix matched to the same NaOH concentration as the samples, there can be significant matrix effects on the apparent Br content of the samples. There is a reference to Balcone-Boissard et al., (2009) for the method, but there needs to be some additional detail given here. Not all the details of the method need to be added, but I would like to know at least some details of the ion chromatography and ICP-MS methods (e.g., what standard concentrations were used, were they matrix matched to the standards for ICP-MS).

Line 606:

Was the V2O₅ heat treated prior to fusion? I know that Balcone-Boissard et al. (2009) indicated that the V2O₅ did not contain significant quantities of halogens, but Segee-Wright et al. (2023) indicated that different V2O₅ batches (even with the same product number and from the same manufacturer) can have starkly different halogen contents, particularly for Br and I. Unless this study used the exact same batch of V2O₅ used by Balcone-Boissard et al. (2009), then there needs to be some concern about contamination from the V2O₅. Line 617 mentions that procedural blanks were run. Did those procedural blanks include V2O₅? This is something that can be quickly addressed.

Lines 613-614:

Here, the names of the rock standards measured need to be reported, as well as the calculated yields are for each halogen. Without this information, it is challenging to assess the quality of this data.

Line 615:

Is this supposed to say <100 g/L? Because this is equivalent to 100,000 µg/mL or 10 wt% F or Cl, which is impossibly high for 0.5 grams of fluxed rock at ~80 mL of solution extracted. I assume this is a typo of some sort. What was this supposed to say?

Line 616:

Is this in the solution or propagated to the rock? Either way, both should be reported for every halogen.

Line 684:

¹⁸O and ³⁰Si were measured on an electron multiplier in serpentines that have high O and Si contents? Wouldn't the counts be extremely high and damage the EM over time? Faraday cups weren't used for measuring those elements? I'm not saying this is wrong, I'm just checking to make sure this is correct because it is different from the volatile measurement procedures I've used at other SIMS facilities.

Lines 689-691:

Show the good correlation between EPMA and SIMS Cl measurements in a supplementary figure.

Lines 690-692:

Was only one standard used to make F and Cl calibration curves? I don't like the idea of using only one calibration standard for Cl and F, particularly when the standard is not matrix matched. This is a problem for both F and Cl, but at least the F contents of most samples are lower than that of the standard, so their values can be interpolated between the standard and the origin. For Cl, the standard has 113 ppm Cl, but the samples extend to 900-1100 ppm Cl. The error associated with extrapolating a calibration curve to 9 times higher than the one standard will result in enormous errors. Do the errors reported for the SIMS data take this extrapolation into account? If they don't, they need to. Why wasn't a higher Cl standard chosen?

Lines 704-707:

Instead of telling the reader that there is a linear relationship, show this in a supplementary figure. This is especially important since these measurements are not routine. If scapolites and glasses of different compositions also fall along a single calibration line, that could be used as an argument against significant matrix effects for these measurements.

Lines 729-737:

Although I do not like that there are no matrix matched standards or that only one calibration standard was run, the reasoning in the section is decent. I don't think the problems I have raised regarding the methods will have a large impact on the main conclusions of this paper.

However, the claim that matrix effects are minimal when comparing two different elements within the same mineral needs to be justified, either through a literature reference or by making the argument in this section. This could be demonstrated by comparing SIMS calibrations of two different elements (F and Cl for example, but it could really be any two elements) in two sets of standard materials with very different matrices that have been previously published. If the ratio of the slope of the Cl calibration line to the slope of the F calibration line is similar for both standard sets, then one could argue that the matrix will have no significant impact on the measured F/Cl ratio.

Even if matrix effects don't change F/Cl, Br/Cl, or I/Cl values by orders of magnitude, matrix effects will almost certainly have some systematic effect on the absolute concentrations of all halogens that will differ between different phases. Although the halogen ratios are unlikely to change enough to alter the interpretations of the manuscript, the concentrations could be systematically offset by >50-100%. There needs to be mention of the deficiencies of the method and the resulting deficiencies of the data in respect to absolute halogen concentrations.

Version 2:

Reviewer comments:

Reviewer #4

(Remarks to the Author)

The authors have adequately addressed my concerns in the revised manuscript and supplement. I recommend that this manuscript for publication.

REVIEWER COMMENTS

Reviewer #1 (Remarks to the Author):

The authors present a detailed mineralogic, petrologic, and geochemical (focused on C and halogens) of variably carbonated serpentinites from Oman (through a drill core to the metamorphic sole). The goal is to determine fluids involved in carbonation reaction, carbonation of the forearc, and the subsequent role of the forearc in the C cycle.

The authors have collected an impressive amount of data linking mineralogy (and different mineral generations) to geochemistry (particularly detailed in situ work). This is no small feat! In fact, determining the different chemistry of the dolomite is key to interpreting the data in context with previously determined geochronology (I thought

this was a particularly nicely done part of the contribution that demonstrated the importance of detailed contextual work).

Thank you for these positive comments and for the many constructive criticisms; addressing these has undoubtedly strengthened and focused the manuscript. We were particularly pleased to read praise for the petrographic and in situ geochemical approach taken.

Major comments:

I recognize that halogen work is extremely difficult, and particularly halogen concentrations via SIMS is limited. Therefore, some more details on methods (see comments below) would be helpful. Also given that some of the same data are collected by different methods, and in some cases have dramatically different results, these differences must be addressed and why some data are used and not others.

Further analytical details for halogen work have been added to the methods along with a section comparing the results between different methods and accounting for any discrepancies. See also detailed responses to individual comments below.

The discussion linking dehydration, carbonation, and depths of these reactions needs to be clarified and better linked to the proposed (2 step) model from the outset.

This section of the discussion (lines 384-457) has been rewritten to more clearly delineate the (admittedly relatively poor) constraints on the PT conditions of carbonation and decarbonation, how these may vary with different subduction zone geotherms, and what may be inferred in the case of Oman. The discussion here has also been reordered to more cleanly divide inferences about the fluid composition (based on the halogen geochemistry and modelling) from discussion about the possible source of such fluid(s). Additionally, clarification has been added to indicate that this section is focused on Stage 1 of the two-stage model (lines 341-345). Finally, a further note has been added to the following section to indicate that implicitly this model suggests Stage 2 must have occurred at similar or shallower pressures than Stage 1 (lines 519-521).

Nature Communications is targeted for a general geology audience. There are numerous places throughout (see comments below) in which a few more details would help the contribution be more accessible. In particular, including other reservoirs (e.g., serpentinites, AOC, mantle) on figures (Fig 7) would be helpful in following the interpretation/discussion.

A large body of literature data (incl. forearc/alpine, seafloor/passive margin serpentinites, AOC, Isua talc-schists/serpentinites, and new annotations) has been added to Figure 7 to better contextualise the halogen analyses within the possible geochemical reservoirs. In addition, numerous changes have been made throughout in

response to detailed comments which have been helpful in making manuscript more accessible and readable (e.g. including Sr isotope ratios).

Minor comments:

Given that Nature Communications allows for 70 references and the authors only have 54 references, many more references can be included particularly in the introductory section. For example, on lines 31-32, the authors state “Recently attention has turned to the potential for hidden outfluxes not sampled by volcanism and unaccounted for in current estimates^{6,7}.” Numerous additional references can be added to this, in which previous authors note commonly overlooked outfluxes via the forearc or backarc, as well as sequestration in volatiles in additional reservoirs (e.g., sub-continental lithospheric mantle). Some suggestions are:

Bekaert DV, Turner SJ, Broadley MW, Barnes JD, Halldórsson SA, Labidi J, Wade J, Walowski KJ, Barry PH. Subduction-driven volatile recycling: A global mass balance. *Annual Review of Earth and Planetary Sciences*. 2021 May 30;49(1):37-70.

Gibson, S. A., & McKenzie, D. (2023). On the role of Earth's lithospheric mantle in global volatile cycles. *Earth and Planetary Science Letters*, 602, 117946.

Barnes, J. D., Manning, C. E., Scambelluri, M., & Selverstone, J. (2018). The behavior of halogens during subduction-zone processes. *The role of halogens in terrestrial and extraterrestrial geochemical processes: Surface, crust, and mantle*, 545-590.

Foley, S. F., & Fischer, T. P. (2017). An essential role for continental rifts and lithosphere in the deep carbon cycle. *Nature Geoscience*, 10(12), 897-902.

Agreed – further reference to the literature have been added throughout the manuscript: this includes several of those listed above (though not all could be included due to additions elsewhere) and more thorough referencing of the geological background literature on the Semail Ophiolite.

Paragraph starting on line 48. At the start of the sentence clarify that you are discussing the listvenites from the Semail Ophiolite. Is this discussing the exact same samples that you studied, or just from the same area?

Prior work was not on the same samples but on samples from the same area (Falk & Kelemen, 2015) and from the same BT1b drill core (de Obeso et al. 2022). This has been clarified as suggested.

“This study focuses on listvenites from Hole BT1b of the Oman Drilling Project¹⁶ drilled in Wadi Mansah (23.364374°N, 58.182693°E) in the Semail Ophiolite. Listvenites from BT1b and nearby exposures show elevated ⁸⁷Sr/⁸⁶Sr ratios” (lines 49-51)

Line 55: Geochronology based on what? More details are needed since this is related to one of the key motivations of the work.

This has been clarified as suggested.

“recent in-situ U-Pb geochronology on samples from localities near (<10 km) Hole BT1b. The latter yielded dates from two dolomite veins and several imprecise dates from listvenites, all <60 Myr, postdating active subduction by at least 20 Myr” (lines 61-64)

Fig 1. Some of the small text in Fig 1a, 1d, and 1e is really hard to read. One could use mineral abbreviations in 1d and 1e to save space, thus allowing for a larger font size.

This is a good point. Fig. 1 has been adjusted as suggested to use mineral abbreviations and increase font sizes for smaller elements.

Fig 1. Show the location of all the samples in the figure, not just those in the reaction zone. Otherwise, it looks like there are only 6 samples.

Sample locations have been added to panel b of Figure 1 as suggested

Fig. 4. d-f and g-i are not labelled on the figure. It looks like it is the same sample in a row, but the caption reads like it is the same sample in a column?

Figure 4 has been redrafted to address comments from Reviewer 3; all panels are now labelled on the new version.

Line 201: Are these EMPA or SIMS data or an average of both?

These are SIMS data. This has been clarified in the text

“In situ data from secondary ionisation mass spectrometry (SIMS) similarly show...” (line 213)

Line 231: Higher F contents than what? It looks like the serpentine from the ophicarbonates has just as high F concentrations as the listvenites. Or do you mean that some listvenites have higher F concentrations than other listvenites?

The meaning was the latter – within the listvenites some carbonates have higher F than others. This sentence has been rewritten to be clearer.

“Carbonates from listvenites are also mostly colinear with serpentine compositions, although a subset show higher F concentrations” (lines 241-242)

Line 238: Since you have the space, you may want to add more references for sedimentary pore fluids. For example:

Muramatsu, Y., Doi, T., Tomaru, H., Fehn, U., Takeuchi, R., and Matsumoto, R. (2007). Halogen concentrations in pore waters and sediments of the Nankai Trough, Japan: Implications for the origin of gas hydrates. *Applied Geochemistry*, 22(3), 534–556.

Given the additions of various other references and since there is already a source for the composition of sedimentary pore fluid there wasn't space to fit this in. I read it with

interest for future work thought, especially as it provides halogen data for solid phases in sediments.

Line 246: Showing some statistics to back up this statement would be helpful (e.g., box and whisker plots for the different samples) because I don't see a trend. Overall, the data look highly scattered. For example, 43z4 has some of the highest and lowest Br/Cl values and some of the highest and lowest I/Cl values.

Indeed, there is a lot of variation between individual spots within a sample but these do show a progressive trend from the least to the most carbonated ophicarbonates. A boxplot of Br/Cl by sample has been added to Fig. 7 to illustrate this more clearly.

Line 250: "seawater serpentinites"- reword. I assume you mean abyssal serpentinites hydrated by seawater? Put serpentinites (both abyssal and forearc) on Fig 7 for reference. "Plot away from" is vague. Be more specific (and adding serpentinites to the figure will help).

A range of endmembers (upper and lower AOC, forearc/alpine serpentinites, abyssal/passive margin serpentinites), none of which provide a convincing match for the low Br/Cl endmember in the BT1b data, have been added as an inset to Fig. 7. It was not possible to include these on the main panel of Fig. 7 without it becoming unreasonably cluttered. Seawater serpentinites was a mistake and should have read seafloor serpentinites (i.e. abyssal or passive margin). The sentence has been rephrased to be clearer:

"The low Br/Cl and I/Cl ratios observed in the ophicarbonates are below the seawater Br/Cl ratio (3.47×10^{-3}) but above seawater I/Cl (3.04×10^{-6}) and show distinctly lower Br/Cl than seafloor or forearc serpentinites^{24,37} (Fig. 7b)" (Lines 259-262)

Line 252: That would mean the serpentine protolith has a very, very low Br/Cl (lower than the reported mantle value of $2.8 \pm 0.8 \times 10^{-3}$; Kendrick et al., 2017). How is this reconciled?

This passage is intended to refer to the halogen composition of the uncarbonated serpentinite prior to carbonation not the fresh peridotite protolith of both. The halogen composition of serpentinite is dominated by that of the serpentinising fluid rather than peridotite protolith except at very low degrees of alteration (e.g. Kendrick et al. 2022 Contrib. Min. Pet.). The use of the word protolith was meant to refer to the protolith for carbonation but we acknowledge that is potentially confusing and the sentence has been rewritten:

"We interpret the correlation between the two ratios as a binary mixing trend between a low I/Cl endmember, representing the uncarbonated serpentinite (or a fluid it has equilibrated with) and a high I/Cl endmember representing the composition of the carbonating fluid." (Lines 262-265)

Line 258: Clarify the arrays. Do you mean samples with high F/Cl and with low F/Cl?

Fig. 6 has been annotated to show these more clearly and the caption clarified: “The data fall into two arrays: one with fairly constant F/Cl (~0.01-1) and one with elevated F and variable F/Cl (1-1000), labelled I and II respectively” (Lines 270-1)

Line 264: A plot of Br vs I (showing different reservoirs) would be helpful.

As this information is implicitly available in Figure 7 (i.e. lines of constant Br/I form linear arrays on this plot) we felt that the discussion in the text was sufficient to get the point across.

Line 270: Add AOC onto Figure 7. AOC span a large range of I/Cl, many with high I/Cl values.

AOC has been added to Fig. 7. Upper AOC would be a more likely fluid source given the proximity of volcanics from the Haybi and Hawasina complexes but this is incapable of supplying the low Br/Cl signature seen.

Fig 7. Given that the focus of the paper is on carbonation, it might be useful to add carbonate to this figure. Since iodate can substitute for carbonate, why is the I/Cl not higher (like as seen in some AOC)?

Based on the coexistence of graphite and haematite, f_{O_2} was relatively low during carbonation ($\log[f_{O_2}] < -40$; Kelemen et al. 2020 JGR:S). This strongly suggests iodine will be speciated as iodide only (e.g. Eh-pH diagram, Fuge and Johnson, 1986, *Env. Geochem. & Health*) which would preclude compatible behaviour of iodate having an influence. Since this is not thought to be a relevant effect we have not included an oxidative carbonate endmember on Figure 7 as doing so would add to an already rather busy figure.

Fig 7. Why are there more whole rock data shown than samples? For example, there are 4 metamorphics listed on Table S1, but more than 4 metamorphic samples shown on Fig 7. Is this because Fig 7 is showing both Pyr and NI-NG-MS data? (If so, then there needs to be some discussion on why the different methods determine different concentrations- see below.)

Where sample material allowed, the whole rock samples were run in duplicate and all these data have been plotted on this figure. This was judged to be the most appropriate approach since the aim is to show the overall trends in comparison to the in-situ data and both intra- and inter-sample are valid measures of this. This has been clarified in the caption: “Bulk data, including duplicate analyses, are shown for listvenites, ophicarbonates and metamorphic samples as inverted triangles and the mean composition for each lithology as large triangles.” (Lines 294-297)

Line 297: The full reference for reference 31 is not given. What are the halogen values for other Oman serpentinites (and their tectonic relationship to your samples)? Can you add these data to your figures for comparison?

Reference has been updated to include full details. These data from four serpentine veins and one whole rock serpentinite are to our knowledge the only halogen data from “normal” Oman mantle section serpentinites. These come from transects through the full ophiolite stratigraphy at Wadi Tayin and Wadi Haymilyah with the mantle section sampled near to the MOHO (0-500m below) and far (>10 km) from the basal thrust (D’Andres, 2021, Figs 2.1, 2.2). These are therefore taken as being representative of the composition of serpentinites in the mantle section (e.g. the “main mantle section” of Godard et al. 2021, JGR:SE), in the absence of carbonation processes.

The field defined by these data has been added to Fig. 7b and the text rewritten to specify the context of these data:

“It is possible that the unusual composition of serpentine in the least carbonated samples reflects an atypical serpentine chemistry characteristic of the Oman mantle section. However, serpentinites sampled far from the basal thrust are more typical of seafloor or forearc serpentinites, with Br/Cl ~close to seawater⁴³ (Fig. 7b).” (Lines 312-315)

Line 304: Since this is key to the presented interpretation, I suggest drawing this trend/arrow on the figure.

Annotated arrow has been added to Fig. 7 as suggested

Line 308: Can you expand on this since it is directly related to your interpretation?

This passage has been expanded upon slightly as suggested: “Notably, a similar trend of correlated Br/Cl and I/Cl, is shown by variably carbonated Isua serpentinites, with low Br/Cl compositions ($>7 \times 10^{-4}$) suggested to arise from fractionation of fluids during water-rock reaction” (Lines 324-326)

Adding those data onto figure 7 (with an arrow depicting the trend) would be helpful. Isua serpentinites and talc-schist have been added to Fig. 7 with an arrow indicating the apparent vector of carbonation for both this study and Isua data.

Line 335: And what is that fluid/rock ratio that is assumed?

Depending on the CO₂ content of the CO₂-rich fluid component, a min. fluid/rock ratio of 1 to 23 is required to explain the fractionation seen. The text has been updated to include this: “the minimum possible fluid/rock ratio required to explain the degree of carbonation observed is assumed (~1 to 23; Supplementary Table S8)” (Line 355-256)

Line 369: Reference?

Missing reference (Kelemen and Manning, 2015) has been added (Line 403)

Line 375: Clarify both scenarios? At shallower and deeper depths? What CO₂ concentrations are used at each of those depths? The assumed parameters are not completely clear. Are these different scenarios related to parts 1 and 2 on figure 5?

This section of the discussion has been substantially redrafted in response to one of the major points about to give a more thorough overview on the constraints around PT conditions of carbonation and decarbonation reactions. We feel this has also clarified the specific point raise here, specifically in this passage:

“For a fluid resulting from dissolution of carbonate with 3 wt.% CO₂ our calculation only yields a solution for very low Cl contents in the carbon rich endmember (Supplementary Table S8). Low salinity waters are found in forearc settings, derived from the breakdown of clay minerals in the slab⁵². However, breakdown of clay with 20 ppm Cl and 7% water³² would be expected to generate fluids with ³⁵300 ppm Cl, significantly in excess of those required by the model. Moreover, final fluid Cl is predicted to be only 35-75 ppm which would be expected to result in very low serpentine Cl contents (cf. mean Cl_{serpentine} = 665 ppm). Fluids from decarbonation reactions (XCO₂ = 0.05–0.2) yield model solutions with higher Cl contents (285-915 ppm) more consistent with those in serpentine. The latter appears to be the more consistent scenario with respect to the available geochemical constraints although we do not rule out high pressure carbonate dissolution.” (Lines 407-418).

Lines 399-440: The volatile release is also going to be a function of the thermal structure of the subduction zone. What is most reasonable for your study area?

This is a good point. This is partly addressed in the redrafted discussion referred to in the previous point, specifically:

“Modelling of devolatilisation reactions in the downgoing slab generally indicate that the main metamorphic release of both CO₂ and H₂O occurs in a pulse between 2.2 and 2.6 GPa, corresponding to depths of ^{51,54,55}60-80 km in the forearc. This would imply lateral flow of fluids into the forearc. However, the depth of decarbonation will also depend on the thermal structure of the subduction zone, with warmer subduction favouring decarbonation, commencing as shallow as 40 km^{55–57}. The Semail subduction zone appears, at least initially, to have been hot as indicated by sediment-derived granitoid melts generated at granulite facies conditions⁵⁸. This may have favoured metamorphic decarbonation at shallower levels. Overall the mixing of sedimentary pore fluid with carbon-rich fluids to produce the carbonating fluid implies lateral migration of fluids, since most pore fluid expulsion occurs relatively shallowly

during subduction^{49,50}. That said, the exact distance between these loci of devolatilisation and carbonation is difficult to constrain and may have been relatively small under a warm subduction geotherm.” (Lines 427-440)

This point is also further discussed under *Implications for subducted carbon fluxes* in response to comments from Reviewer 3. See lines 546-591 and Supplementary Text S2.

Line 403: Based on the discussion starting on lines 368, it is not clear exactly what the depths of carbonation are. This discussion linking dehydration, carbonation, and depths of these reactions should be streamlined and better linked to the steps outlined in the model later in the manuscript.

We agree this was a deficiency in the original draft. This section of the discussion has been substantially rewritten to streamline and clarify things as suggested. See comments immediately above and in response to major point.

Relevant changes too long to reproduce here, see lines 377-445.

Lines 404-405: This sentence seems like an add on.

Agreed, sentence was removed during rewriting

Line 418: delete “etc” and add “e.g.,” before dolomite

Amended as suggested

Line 462: Careful with the wording here because the serpentinites in this study do not have halogen ratios similar to those from other forearc regions (other regions have higher Br/Cl and I/Cl plotting up in the pore fluid range). (The subsequent explanation on the following lines on the evolving composition is much clearer here than previously in the manuscript. Some of these concepts need to be better introduced earlier to avoid confusion.)

Agreed – the BT1b serpentinites don’t show evidence for a forearc setting for serpentinitisation. This passage has been rephrased.

“Rather than a single event we envisage formation of the listvenites as a multi-stage process: prior to carbonation the mantle peridotite protolith may have been partially serpentinitised in a seafloor setting⁴³. Subsequently the BT1b mantle section was incorporated into the forearc of a young subduction zone.” (Lines 498-501)

Line 478: Reference? This is interesting, but if going to be mentioned then should be introduced earlier in the discussion.

This has been included as a possible driving force for carbonation in the introduction: “in the latter it is a product of some late-stage tectono-magmatic event in Oman’s

geological history (e.g. elevated heatflow related to minor Paleogene magmatism²³) and of regional rather than global significance.” (Lines 66-69)

Line 508: Define Prop CIPF

This sentence has been rephrased to be clearer: “Mass balance of the composition of the carbonating fluid, assuming the proportion of Cl from pore fluid (48-87%), a pore fluid chlorinity of 0.11-0.15 wt.% and pore-fluid flux of 1×10^{13} kg/yr in the forearc⁶³, yields a deep Cl flux of 4.5×10^{10} – 1.6×10^{11} g/yr.” (Lines 567-569)

Line 550: A few more details on the bulk rock pyrohydrolysis + IC/ICP-MS halogen concentrations are needed. For example, what are the yields (or concentrations) of halogens on rock standards (e.g., Kendrick et al, 2018)? What are the blanks (e.g., halogen blanks from NaOH, V₂O₅, etc.)? Are blanks corrected for or are they all below detection limits? What were the analytical protocols for the IC and ICP-MS- or at least a reference to where these are outlined?

Further details of the method have been added and the reference for the full analytical protocols for IC and ICP-MS (Balcone-Boissard et al. 2009). Yields are difficult to distinguish from other sources of error (e.g. inhomogeneity of standards for halogens, analytical error) but typical uncertainties have been added in the text. Blanks were not subtracted as they were always below detection limits. This has been clarified in the methods.

“A split of whole rock powders was analysed for bulk halogens (F, Cl, Br, I) at Institut de Physique du Globe de Paris, (IPGP), Paris, France. Full details of method and analytical procedures are given in [69]. [...] Repeat analyses of rock standards indicate mean relative errors of ¹⁸10% for F and Cl, ⁸¹30% for Br and ¹²⁷60% for I [69]. Detection limits were well below sample concentrations for F and Cl (<100 g/L) and were approximately 10ppb and 1ppb for Br and I, respectively. No blank correction was necessary as procedural blanks were always below detection limits.” (Lines 602-604, 612-616)

Line 584: Were rock standards analyzed via NI-NG-MS? Comments on the reproducibility of samples analyzed via NI-NG-MS? Given the poor reproducibility of Br and I on some samples via pyrohydrolysis and ICP-MS and the explanation was “sample heterogeneity or less reliable analytical reproduction of Br and I” it would be good to see if potential sample heterogeneity is also documented via NI-NG-MS.

A comparison of Br and I concentration data collected by the two different methods shows stark differences. The Br concentrations via pyrohydrolysis are approximately 50% less than those of the noble gas method, except for one sample with ⁸¹470% more Br. The same is seen with I, in which I by pyrohydrolysis records much lower concentrations except for a couple with ¹²⁷400-500% more I. Some mention of the poor comparison is needed.

Secondary rock standards were not analysed however several primary standards and flux monitors are included in each irradiation tube and the results are reported relative to those standards (see Kendrick, 2012 Chem Geol, and Ruzie-Hamilton et al. 2016 Chem Geol, for detailed discussion of the method). The main reason secondary standards are not included is the general lack of well-characterised homogeneous and matrix matched standards (e.g. for serpentine). A further issue is the limited space in an irradiation tube and the large proportion of the tube taken up by primary standards/monitors. Given the significant costs associated with irradiation, adding secondary standards too would be prohibitive.

This approach is common across the labs engaged in halogen analyses of these type (see e.g. papers from Kendrick's group). We also note that similar issues have been faced trying to identify rock standards for Ar-Ar with inhomogeneity of potential standard materials a common pitfall (e.g. Alexander and Davis, 1974, GCA) and these too are commonly reported without additional secondary rock standards.

Regarding comparison of datasets, a new section has been added to the methods to specifically comment on the agreement or lack thereof between different halogen analyses (pyrohydrolysis, NI-NG-MS, SIMS, EPMA) and explain the choice of NI-NG-MS over pyrohydrolysis data (in response to comment below).

We note that there is good agreement between SIMS and NI-NG-MS results and suggest that low abundance of Br and I in pyrohydrolysis may be due to low yields:

“Comparison of bulk halogen data collected by NI-NG-MS and pyrohydrolysis reveals some notable differences. Chlorine abundances generally agree reasonably well however, the abundance of Br and I is lower in most pyrohydrolysis analyses suggesting low yields during pyrohydrolytical extraction, possibly related to the unusual sample matrix. Monitoring sample-by-sample yield requires the use of hazardous radioisotope tracers (e.g. ^{125}I [76]) and was not performed in this instance, so this cannot be confirmed.

It is difficult to directly compare SIMS and NI-NG-MS data since these were measured on individual minerals and bulk rocks chips, respectively. Nonetheless, the Br/Cl and I/Cl ratios are all within error of the mean of SIMS data from serpentine (which is likely to dominate the whole rock signature). Moreover, bulk NI-NG-MS data and individual serpentine SIMS data are colinear in Br/Cl – I/Cl space suggesting any slight differences arise due to differences in the proportions of different serpentine endmember compositions analysed by SIMS versus the bulk composition of the sample. The SIMS chlorine data also agree well with results from EPMA. The correspondence of halogen data from NI-NG-MS, SIMS and EPMA shows the consistency of the data across multiple methods.” (Lines 738-753)

In Table S2 it says that Cl, Br, and I values determined by NI-NC-MS are preferred. Why? But, then in Table S1, it says that F and F/Cl uses F and Cl data measured by Pyr-IC. Why is the Cl measured by Pyr-IC used if NI-NC-MS is preferred? And is the Cl Pyr-IC data only used for the F/Cl and not [Cl]? In Table S2, it is denoted that Cl is measured via ICP-MS (not IC).

This is now explicitly discussed in an additional section to the method referred to immediately above. In short, there were a large number of samples below detection for Br and I by pyrohydrolysis which made it complicated to calculate a meaningful mean composition. Producing a complete whole rock dataset for these elements was the main motivation for the NI-NC-MS analyses.

F/Cl was calculated using Cl from pyrohydrolysis because of the potential for intrasample heterogeneity, meaning a ratio of two analyses made on same sample aliquot is likely to be more accurate.

“A significant proportion of listvenite samples had Br and I below detection by pyrohydrolysis. This makes calculation of a mean composition problematic and prone to systematic error depending on the assumptions made about the distribution of the data (e.g. log-normal, gaussian, uniform). In producing an overall bulk halogen dataset, the NI-NC-MS data are preferred for the heavy halogens (Cl, Br, I) since they provide a complete dataset for all samples analysed. These data are supplemented by the F abundance from pyrohydrolysis. To avoid any potential inaccuracy due to analytical differences between the two methods, the F/Cl ratio for each sample was calculated using the Cl abundance measured by pyrohydrolysis.” (Lines 754-762)

Table S2 – this was an error. Cl was measured by IC. The table has been corrected

Line 617: Was only one glass standard used for Cl and F SIMS analyses? Is there concern about matrix effects given that serpentine, carbonate, and vein material was analyzed? On Table S5, both EMPA and SIMS Cl data are given. With the exception of some very high Cl concentrations via EMPA, there is reasonable agreement. This would be good to point out.

Yes indeed matrix effects are a potential concern however this is difficult to address as there simply aren't matrix matched standards available. This would certainly be an area of interest to improve the method.

However, we do not believe this affects our conclusion. The most important interpretations are based on a) ratios (e.g. F/Cl) where matrix effects are likely to at least partly cancel out and b) on relative differences between samples of the same matrix (e.g. comparison of carbonate compositions). For instance, although there might be some difference in matrix effect between magnesite and dolomite this is unlikely to result in a 100x increase in the apparent abundance of F in the later. Moreover, we see

some dolomite with low F and some magnesite with high F (Fig. 6) clearly indicating that matrix effects alone cannot explain the variation seen.

As well as the good agreement between Cl measured by SIMS and EPMA, we also note the good agreement between NI-NG-MS and SIMS data for serpentine (see Fig. 7) wherein all SIMS data and bulk ophicarbonates data lie colinear in Br/Cl – I/Cl space. Given the large degree of scatter which typifies most heavy halogen datasets, even from a single locality (e.g. Fig. 7b; Carter et al. 2021 GCA) this array is strikingly tightly defined and gives us good confidence in the robustness of both datasets.

The potential for matrix effects is now explicitly commented on in the methods:

“Matrix effects are a potential issue in SIMS analysis⁷⁵. These effects are difficult to assess or correct for since there are no widely available matrix-matched standards available for halogens in serpentine or magnesite. Nonetheless, the good agreement of SIMS, EPMA and NI-NG-MS data (see below) lends confidence that any matrix effects are relatively minor. Furthermore, differences in matrix effects are likely to be minimised, although not absent, in analyses of the same mineral, meaning that relative differences in compositions of a given mineral can be compared with some confidence.” (Lines 728-734)

Table S1: What does “shallowater-corrected data used for I abundance and I/Cl ratios” mean?

The production of $^{128}\text{Xe}_I$ from I during irradiation can be monitored using either scapolite mineral standards (which are also used to monitor $^{80,82}\text{Kr}_{Br}$ production from Br) or from Shallowater meteorite irradiated in the tube with the samples. Generally, Shallowater-corrected data is preferred as it has lower intrinsic uncertainty, although the two values are almost always within uncertainty of one-another (and indeed offer a helpful check on the internal consistency of the data). Both values can be compared in Table S5.

Reviewer #2 (Remarks to the Author):

This manuscript focuses on the carbon cycle in subduction zones, exploring the role of carbonated mantle peridotites in sequestering subducted CO₂ through a study of the Oman ophiolite. The research is of great significance, aiming to address the long-standing debate regarding the origin of carbonating fluids and their impact on the global carbon cycle. The approach combines multiple analytical techniques, which is innovative and promising. In general, I think this manuscript is well written. I don't have special comments for the authors. I would suggest acceptance for this manuscript.

Thank you for the positive comments it is nice to read that you found the approach innovative and the manuscript well-written.

Reviewer #3 (Remarks to the Author):

Carter et al. present a well written and illustrated paper integrating careful petrography with multiple geochemical techniques, several of which are innovative in their application to the topic. The paper presents a new, untapped line of evidence (halogens) and a compelling petrographic case for a two stage listvenite formation in the SE Oman ophiolite. This conclusion does the best job so far in the literature of reconciling the controversial mismatch between previous geochronological and geochemical interpretations. The authors take this conclusion and point out its implications for a hidden carbon sink in forearc mantle, which would represent an important component of subduction systems and the deep C cycle.

These are the now the most studied listvenites in the world and will doubtless continue to receive attention from the subduction zone cycling community if a forearc wedge carbonation interpretation is partly true. The rocks and reactions are also highly relevant to recent efforts to engineer peridotite carbonation as a means to permanently sequester CO₂. Halogens as important fingerprints, ligands, and catalysts in such reactions are an important but overlooked part of this effort.

The subject matter is of vibrant interest and the conclusions are of relevance to several different swaths of our community: those working on listvenites and ophiolite, those on subduction zone cycling, and those on global biogeochemical cycling.

Thank you for your thoughtful comments, both positive and constructively critical. These have been addressed in full, and this has certainly improved the manuscript. We were particularly pleased to read the positive comments regarding our attempts to reconcile of geochemistry and geochronology.

I have a few concerns below and many minor comments, that should hopefully be possible to address with minor/moderate revisions.

General comments

The main argument I see against the principle thrust of the paper – that forearcs are a hidden carbon sink, is that listvenites are discontinuous along the metamorphic sole in Oman, and rarely present in other ophiolites with exposed soles (?), and have not been

dredged from forearc sections in the Marianas or Tonga trenches. Can this be considered, explained or conclusions tempered to reflect this reality? Showing the locations of all the listvenites described in the ophiolite in a more detailed Fig 1a would assist this discussion.

This is an important comment, and we accept that there is more work to be done to fully delineate the controls on listvenite formation and whether this might be favoured in certain subduction zones or at certain times in geological history due to factors such as thermal structure, sedimentary input. In particular we note that the Marianas and Tonga forearcs while probably the best studied in terms of dredges and drilling into mud volcanoes also may have poor potential for listvenite carbonation since they have low carbonate content in sedimentary input (e.g. GLOSS-II; Plank, 2014) and cold geotherms neither of which favour generation of CO₂-rich fluids, especially at relatively shallow depths (e.g. <50km). Production of CO₂-rich fluids is favoured by hot subduction geotherms and mixed carbonate-siliciclastic sediment input (Stewart and Ague, 2020, Nat. Comms.) both which appear to have been true of Oman (e.g. production of plagiogranites by melting of Haybi complex, Cox et al. 1999, Contrib. Min. Pet.). Where CO₂-rich fluids are allowed to migrate without re-equilibration (e.g. channelised along the slab interface) they may promote significant carbonation. It is also notable that the Central American forearc which shows strong evidence for carbonation occurring at depth (Barry et al., 2019, Nature) is subducting sediments of intermediate composition (~25% CO₂; Plank, 2014).

The slight dichotomy between evidence from Oman and well-studied forearc is now included in the final section of the discussion and discussed in detail in Supplementary Text S2. The conclusions have been tempered somewhat to reflect uncertainty regarding the precise conditions which may favour listvenite formation in forearcs and, importantly, the potential for these to have varied through Earth's history.

“However, it should be noted that listvenites have not been directly observed in dredges or drilling at modern forearcs such as Mariana and Tonga which allow fluid compositions at slab depths to be inferred. This highlights key differences between those and the Semail subduction zone which may promote listvenites genesis in the latter, including: subduction geotherm (cold⁶⁵ vs hot²⁹), sedimentary input (dominantly siliceous, 2.6% CO₂⁶⁶ vs mixed volcanic-carbonate-siliciclastic, 8.6% CO₂-^{18,67}), and even redox state and the relative timing of serpentinisation and carbonation⁶⁸ (see Supplementary Text S2). This raises the important possibility that the magnitude of the forearc carbon sink may vary spatially and temporally in response to changes in oceanic sedimentation, subduction zone thermal regimes and redox state over geological time and could therefore play an important role in fluctuations in the long-term carbon cycle.” (Lines 581-591)

Listvenites are widely distributed throughout the Semail ophiolite near to the basal thrust. We also note that the metamorphic sole is also discontinuously exposed along the basal thrust having been removed by late-stage brittle thrusting (Searle et al. 2022, *J. Struc. Geol.*). It is not unreasonable to imagine a similar fate for listvenites both in Oman and other ophiolite localities. Finally, although listvenites are not present in all SSZ ophiolites they are nonetheless relatively common as demonstrated by the recent review of Menzel et al. (2024; *Earth Sci. Rev.*). The lack of listvenites in some ophiolites may relate to poor preservation potential during obduction or to differences in formation potential as discussed above; more work is needed to clarify these important points.

“Listvenites are widespread although not continuously exposed throughout the ophiolite (Fig. 1a). The metamorphic sole is also not ubiquitously preserved and in numerous places late brittle thrusting has juxtaposed relatively unmetamorphosed allochthonous sediments against the mantle section³¹. This suggests that listvenites might also be prone to removal during obduction and may therefore have relatively poor preservation potential, especially in less laterally extensive ophiolites. In the broader geological record there are numerous examples of listvenites associated with the basal thrusts of supra-subduction zone (SSZ) ophiolites although often with some ambiguity as to the timing of their formation relative to obduction¹¹. Nonetheless they are not ubiquitous in SSZ ophiolites with exposed metamorphic soles which suggests either that they are poorly preserved or that specific conditions are required for their genesis.” (Lines 549-560)

Fig. 1 has been updated to include published listvenite localities (Nasir et al., 2007, *Geochem.*; Boudier and Nicolas, 2018, *Tectonics*).

Its important to note that Scharf et al. worked on listvenites from another locality, and that this study focuses only on listvenites from BT1B. While the drilled localituy is the among the largest listvenite bodies in the ophiolite, it is not the only one. If there were indeed two pulses of carbonating fluids, then it would follow that they have different spatial foci/extents, which might explain differing conclusions from workers in different parts of the ophiolite.

Agreed – the key passage has been updated to explicitly suggest this as a reason for divergence of opinion in the literature:

“Overall these results are unsurprising if ophiolitic mantle sections are viewed as a patchwork of overprinting metasomatic events (e.g. ref 24) not all of which may be apparent or detectable at all scales or at all localities. It is likely that multiple stages of fluid infiltration affected the ophiolite prior to, during and since obduction and may have overprinted some chemical or isotopic systems. Furthermore, the extent of such overprinting may have varied spatially resulting in different conclusions from studies at

different sites (e.g., ref. [17] cf. [21]). Nevertheless, our data indicate a strong subduction affinity which has escaped such overprinting and attest to fluxing and carbonation of the shallow forearc by CO₂-rich slab derived fluids.” (Lines 523-531)

The conclusions of this paper rely heavily on the petrographic evidence for dolomite overprinting magnesite. I don't find the evidence presented very clear/convincing. Can it be expanded on or shown in higher resolution/different imaging types?

We accept that the original version of Figure 4 didn't show the petrographic evidence in high enough resolution. This figure has been redrafted with 4 insets added showing key areas at higher magnification and annotated with our interpretation of cross-cutting relationships. We have also highlighted in the text that many other studies have interpreted dolomite as a late forming phase.

“Our observation that dolomite is a late-forming phase is corroborated by numerous other studies of listvenites from BT1b and nearby localities^{19,33,35}.” (Lines 479-481)

The final panel of Fig 8c and the accompanying metamorphic sole F-rich fluid source interpretation doesn't make sense. By the time of this F rich dol precipitation (~60 Ma by this model), the sole had already been metamorphosed to amphibolite facies (and exhumed). Exhumation and retrogression of these rocks cannot produce a fluid at shallow depths – they are almost dry following their trip into the mantle. The source must be tectonically deeper than the sole in lower grade rocks. There are also several nappes beneath the ophiolite that are omitted here, as 'Arabian continent margin' evokes the autochthonous continental basement. I recommend consulting the cross sections in Searle (2007, Georabia) to get the tectonostratigraphy correct, and this may help with your interpretation of F rich fluid source options.

We agree that the original version of this figure was potentially misleading. The discussion and figure have been updated to better reflect the relative timing of events and geological relationships. We suggest the F-rich Stage 2 fluids may have derived either from metamorphism of underlying shelf carbonates during obduction (e.g. Grobe et al. 2019, Solid Earth) or from fluid circulation associated with Paleogene magmatism (e.g. Wilde et al. 2002, J. Virtual Explor.; Scharf et al. 2020, J. Afr. Earth Sci.). Their similarity in composition to the metamorphic sole lithologies is suggested to derive from interaction with allochthonous sediments/volcanics from which the metamorphic sole formed although it could be co-incidental (e.g. reflecting an F-rich magmatic fluid source).

Discussion has been redrafted to more clearly explain proposed model:

“Later, Stage 2 dolomite and magnesite crystallisation occurred from a Ca-rich fluid with higher F/Cl and I/Cl ratios. This stage is only dominant in some horizons and

microstructurally crosscuts earlier carbonates. Fluids derived from low temperature metamorphism of autochthonous shelf carbonates during obduction⁵³ are a plausible source for the F- and I-rich Stage 2 fluids, especially given the marked enrichment of Ca in the lowermost listvenites overlying the basal thrust²⁰. Their similarity in halogen composition to the metamorphic sole may relate to interaction with overlying allochthonous units, the protoliths for the metamorphic sole^{20,31}. Equally, Stage 2 carbonates could be derived from interaction with another F-rich reservoir (e.g. fluid circulation associated with later igneous activity^{23,62}).” (Lines 511-520)

Figure 8 has been redrafted to clearly show the nature of underlying autochthonous and allochthonous units and to show more explicitly the two possible formation mechanisms for Stage 2 carbonation.

Figure comments:

Fig. 1. The intervening met. sole close up section (Fig 1c) breaks the flow of the figure from left to right. Is it really necessary considering it has been well described and no samples in this study come from it? If you deem it so, it may be better to either side of the full 0–300 m stratigraphy.

Fig 1c (now Fig. 1e) does not show the metamorphic sole but the reaction zone between listvenite and serpentinite (~100 m depth) which this manuscript focuses on. We do accept, however, that its position broke the flow from Stratigraphy to Mineralogy. This has now been moved to rightmost.

Fig. 2. Annotate the side of the fig. with an arrow indicating carbonation intensity.

Annotation has been added as suggested

Fig. 4. I don't find the dol x-cutting particularly convincing, at least at the resolution in the merged PDF. Dol doesn't have euhedral faces, it looks like the Mgs is corroding early dol. The red vs white arrows don't seem to have differing significance?

The first formed magnesite veins have a Fe-rich medial lines and grade outwards to Fe-poor magnesite. This structure can also be seen in Sample 44z3 (Fig. 2b). This structure is clearly crosscut in numerous places by dolomite. Dolomite also appears to be growing interstitially between magnesite with globular and lozenge like habits (Fig. 4f). We are not the first to observe that dolomite forms as a late phase really to most magnesite-dominated carbonation; Beinlich et al. (2020, JGR:SE), Menzel et al. (2022, Solid Earth) and Decrausaz (2023, Eur. J. Min.) all note the formation of dolomite principally as a late phase in BT1b listvenites (and 2 further sites in the latter reference). Menzel et al. (2022) likewise note the common occurrence of Fe-rich medial lines in the

earliest forming magnesite. These indicate that our petrographic observations are robust and are confirmed in study of a wide-range of samples from BT1b (and beyond).

With the exception of Menzel et al. (2022), these studies were already cited in the original manuscript:

“Textural evidence from this and other studies indicates that dolomite is a late forming phase^{19,33–35}; it primarily occurs either overgrowing/crosscutting zoned magnesite-ferromagnesite veins or as an interstitial phase (Fig. 4).” (Lines 184-187)

Very few of the carbonate phases show idiomorphic or euhedral forms, and have been taken to suggest formation under disequilibrium conditions (Beinlich et al. 2020). However, some of the dolomite does show subhedral lozenge-like habit which may reflect free growth into porosity following a phase of dissolution.

The red and white arrows in Fig. 4 highlight dolomite cross-cutting and growing interstitially between magnesite crystals respectively. This is stated in the figure caption.

How closely does the drill hole depth approximate the true pseudostratigraphic depth? This should be mentioned in the caption or annotation.

Hole BT1b was drilled at an incline so depth in Hole is close to true pseudostratigraphic depth. Caption to Fig. 1 has been updated to reflect this:

“The hole was drilled sub-perpendicular to lithological boundaries so depth is approximately true thickness¹⁶.” (Lines 93-94)

Fig. 8. IN panel a, it looks like the mantle wedge is dehydrating and decarbonating.

Agreed. Figure has been amended to be clearer that sediments are dehydrating

Line by line comments:

L45. The tectonic setting of the ophiolite does still attract controversy, a reference here would serve to illustrate which of the various models the authors prefer to contextualize this work.

A reference (Goodenough et al., 2014) has been added as suggested (Line 46)

L50. Adding specifics of the Sr ratios in parentheses would be helpful here. Also, for transparency, note that they overlap the Hawasina metasediments, but also other crustal materials present in the sub-ophiolitic sequence, i.e. they are also likely intermediate between the protolith and the more radiogenic autochthonous basement.

Specifics of Sr isotope ranges mentioned have been added as suggested. However, we are not aware of any work which suggests involvement of autochthonous basement in

listvenite formation, we therefore feel that reference to this would be somewhat distracting and potentially misleading.

“This study focuses on listvenites from Hole BT1b of the Oman Drilling Project¹⁶ drilled in Wadi Mansah (23.364374°N, 58.182693°E) in the Semail Ophiolite. Listvenites from BT1b and nearby exposures show elevated $^{87}\text{Sr}/^{86}\text{Sr}$ ratios (0.7092–0.7145) relative to uncarbonated peridotites ($^{87}\text{Sr}/^{86}\text{Sr}$ 0.7028), Cretaceous to modern seawater or groundwater ($^{87}\text{Sr}/^{86}\text{Sr}$ 0.707–0.709), but which overlap the isotopic composition of underlying allochthonous Hawasina metasediments (0.7082–0.7241)^{17,18}.” (Lines 49-54)

L55. There is certainly a mismatch in the interpretations of the trace elements and geochronology, the key point to made/tone to be struck is that the uncertainty/apparent nature of the dichotomy is not a fault of the data, but interpretation of them on one or both sides of the argument. Set out the timing constraints specifically here, as it is crucial context.

This is a fair point. The sentence has been slightly rephrased to put the emphasis on the interpretation rather than the data. The timing constraints have also been set out in more detail:

“However, there appears to be a mismatch between interpretations based on trace element and isotope geochemistry which suggest a forearc subduction zone setting for listvenite formation^{17,18,20,21} and those from recent in-situ U-Pb geochronology on samples from localities near (<10 km) Hole BT1b. The latter yielded dates from two dolomite veins and several imprecise dates from listvenites, all <60 Myr, postdating active subduction by at least 20 Myr²².” (Lines 59-64)

L63. State which halogens.

Amended as suggested: “The heavy halogens (Cl, Br, I) are hydrophilic and make ideal tracers for interrogating fluid processes in ophiolites^{24–26}” (Lines 71-73)

L78. What other volatile cycles does it have a bearing on?

Amended to include examples: “may be a significant and overlooked part of the global carbon and other volatile element cycles (e.g., H₂O, Cl).” (Lines 84-85)

L94. The origin of those metasediments/metabasalts is of importance for context here, as is the intervening deeply subducted and exhumed met. sole. See Searle’s papers for reference.

Further geological context for units underlying basal thrust has been added as suggested:

“The basal thrust of the ophiolite tectonically juxtaposes mantle peridotites on top of allochthonous metasediments and metabasalts of the Hawasina and Haybi complexes which are themselves underlain by autochthonous shelf carbonates^{29,30}. A

greenschist to amphibolite grade metamorphic sole formed by subduction and exhumation of allochthonous lithologies is discontinuously exposed along the basal thrust-30,31.” (Lines 108-113)

L96. Reference for the Oman Drilling Project?

Reference (Kelemen et al. 2020, Proc. Oman Drilling Project) added as suggested (Line 116)

L97-101. *More tectonostratigraphically complete. This sequence is not a stratigraphy. Tectonostratigraphy/pseudostratigraphy are sometimes used in this situation. Or otherwise avoid the term altogether.

Accepted. The term has been omitted here and replaced with pseudostratigraphy where necessary.

L107. How is the minimal impact of protolith heterogeneity established for these rocks? Sharp compositional variations between dunite/harzburgite are common near the sole.

We accept this point. Such protolith heterogeneity is very unlikely to impact the halogens which will have low concentrations. Nonetheless we don't have a strong constraint on the degree of protolith heterogeneity and so reference to this has been omitted:

“This interval shows a sharp compositional gradient over which the primary mineralogical variation is due to the degree of interaction with the carbonating fluid.” (Lines 123-125)

L110. Reaction zone at the base of the upper listvenite? Or where?

The reaction zone is near the base of the upper listvenite but within it (Fig. 1b). The boundary between upper and lower listvenite is made on the basis of trace element geochemistry at 110 m depth (Godard et al., 2021, JGR:SE). The original version of Figure 1 was based on the original shipboard logging. Fig. 1 has now been updated to align with the geochemical division of Godard et al. (2021) and clearly shows the location of the reaction zone.

The depth of the serpentinite body in question is now indicated in the text to clarify this: “Within the upper listvenite a 20 m thick lens of serpentinite and ophicarbonates (partially-carbonated serpentinite) is preserved (80-100 m depth) and affords almost complete preservation across a 5 m thick reaction front between the serpentinite protolith and fully-carbonated listvenite (Fig. 1e).” (Lines 120-123)

L133. Along the reaction zone? This needs clarification (further into the reaction zone? With increasing carbonation?)

Clarified as suggested: “Serpentine mineral stoichiometry from electron probe microanalysis (EPMA) indicates variable mixtures of lizardite with increasing proportions of talc with increasing carbonation” (Lines 147-149)

L132. Don't all three serpentine minerals have similar stoichiometry?

Yes indeed. Lizardite and chrysotile are true polymorphs, antigorite has very slightly higher Si content. Based on petrographic observations, mesh textured lizardite is the main serpentine phase present (cf. bladed forms typical of antigorite, e.g. Carter et al. 2021, GCA). Similar observations were made by Beinlich et al. (2020, JGR:SE). The main point being made here is that (whatever the polymorph) the serpentine is accompanied by variable proportions of talc in a microcrystalline mixture reflecting variable extent of carbonation.

Some extra detail of petrographic observations added to preceding paragraph:

“Petrographic observations and mineralogical analysis by powder X-ray diffraction (XRD) reveal that ophicarbonates consist predominantly of mesh-textured lizardite with variable replacement by carbonate and talc (Fig. 1c, d, Fig. 2).” (Lines 145-147)

L217-219. Would it? This is not a closed system process, the reaction proceeds because fresh CO₂ rich fluids are introduced and spent ones (necessarily) removed, maintaining high CO₂ activity and high fluid/rock ratios. The implication would be Cl addition to serpentinites downstream. Is this seen?

This isn't necessarily an instantaneous process but incrementally yes: each time some proportion of serpentinite is converted to talc or carbonate it takes in CO₂ and expels Cl which must be lost to the fluid. There is potentially some petrographic evidence of this in the form of Cl-rich veins which increase in abundance with carbonation intensity (e.g. Fig. 2).

Indeed, we would expect to see Cl increase “downstream”. It is hard to assess if this is happening – it could have contributed to the higher Cl contents of sample 44z1 (the least carbonated). However, more detailed sampling extending further from the front of listvenite formation would be necessary to really map out the presence of such an effect.

L296. It is not surprising the Oman serpentinites have markedly different halogen compositions to seafloor serpentinites. Can a more relevant comparison be made to other ophiolitic serpentinites that would rule out the distinct possibility this is some common feature of deeply formed, ophiolitic serpentinites (or those formed by meteoric fluids?)?

A direct comparison of the reaction zone serpentine with seafloor, forearc and ophiolitic serpentinites has been added. However, it does in fact it appears that serpentinites in

Oman far from the basal thrust have similar halogen compositions to seafloor serpentinites (D'Andres, 2021, thesis; Fig. 7).

“It is possible that the unusual composition of serpentine in the least carbonated samples reflects an atypical serpentine chemistry characteristic of the Oman mantle section. However, serpentinites sampled far from the basal thrust are more typical of seafloor or forearc serpentinites, with Br/Cl close to seawater⁴³ (Fig. 7b).” (Lines 312-315)

L306. What is meant by equilibration of a serpentine protolith? Recrystallization?

Chemical exchange could occur in the solid state but admittedly this might be quite inefficient at low temperatures and with low porosities. Recrystallisation is another possibility, especially given the abundant mineralogical transformations the rocks have undergone. Given the pseudomorphic preservation of mesh texture even in fully carbonated rocks, this may not have left clear texture evidence.

Passage has been rephrased: “either by equilibration/recrystallisation of serpentine with the fractionated fluid or by direct serpentinisation” (Lines 322-323)

L407. It is more than ‘somewhat’ in conflict, at least for a single event carbonation model, which is worth specifying. Also, it should be ordered 100–80 Ma; however, recent petrochronological work from e.g. Garber et al. has failed to reproduce the >100 Ma Guilmette et al prograde dates, so it may be safer to say from ‘at least 96 Ma to “80 Ma’ or to focus on the cessation date, which is the relevant one here.

We agree with both suggestions. Passage has been rephrased:

“However, despite this geochemical evidence requiring input of decarbonation-derived CO₂-rich fluids, a subduction origin for carbonating fluids is contradictory to interpretations from recent geochronology which has dated dolomite veins in Oman listvenites to 60 Ma and younger²², significantly after the cessation of subduction at “80 Ma 59,60.” (Lines 441-445)

L441. I agree this is significant, but the petrographic evidence as provided is not that clear/definitive. In line with my previous comment can this be bolstered?

See comments above in response to major point. Although they are mentioned earlier, we have added another reference here to other studies which corroborate late dolomite formation:

“The dating of Scharf et al. 22 relies primarily on analyses of dolomite. Our observation that dolomite is a late-forming phase is corroborated by numerous other studies of listvenites from BT1b and nearby localities^{19,33,35}.” (Lines 478-481)

L446. These samples were from a different locality, so I would not say that they have been identified (with certainty implied) as belonging to a distinct carbonation event. Your data rather provides a compelling explanation as to why they may belong to a distinct carbonation event and potentially represent minimum ages.

This is fair point. The passage has been slightly rewritten to highlight the difference in locality between the studies and to more appropriately qualify the uncertainty associated with extrapolating results between sites:

“The dating of Scharf et al. 22 relies primarily on analyses of dolomite. Our observation that dolomite is a late-forming phase is corroborated by numerous other studies of listvenites from BT1b and nearby localities^{19,33,35}. The identification of BT1b dolomites as belonging to a distinct carbonation event with a markedly different fluid composition therefore suggests that the young dates from other sites may represent minimum ages for the main phase (Stage 1) of carbonation, albeit with some inherent uncertainty in extrapolating results between localities. This would suggest that the bulk of carbonation occurred before 60 Ma, coincident with subduction, as indicated by a range of geochemical proxies – not least the sedimentary pore fluid halogen signature and carbon isotopes¹⁸ of CO₂ bearing fluids.” (Lines 478-488)

L451. It should be noted that this statement is limited in scope to BT1B, clearly some aspect of the listvenites (primary or secondary) was suitable to dating to Scharf et al.

We have added a qualifying that we are discussing the bulk geochemistry of BT1b listvenites and qualified the final statement with the proviso that it may not apply if bulk geochemistry of the samples dated by Scharf et al. (2022) differs significantly from that at BT1b. There is not to our knowledge any bulk geochemical data available for these samples. However, there is no particular reason to believe that these listvenites are of completely different (i.e. high U/Pb) composition and so we feel that this is a reasonable statement to make given the available evidence.

“It should also be noted that the bulk chemical characteristics of the BT1b listvenites suggest that they are highly unsuitable for U-Pb dating. All the listvenites and associated serpentinites have very low U/Pb ratios and lie at the extreme end of a trend defined by forearc serpentinite compositions reflecting low fluid U contents under low fO₂ conditions^{20,61}. A priori it would be expected that any attempt to date such materials by U-Pb methods would be inherently biased towards later perturbations under more oxic conditions where U would be more soluble in fluids. We tentatively suggest that, unless their listvenite bulk geochemistry differs significantly from that at BT1b, a sequence of later fluid infiltration events may be what was dated by Scharf et al.²².” (Lines 489-497)

L460. Is this parallel list not reversed?

Yes indeed this was a mistake and has been amended:

“Water- and carbon- rich fluids derived, respectively, from sediment compaction and decarbonation or dissolution reactions deeper in the subduction zone reacted with the serpentinites forming talc and eventually carbonate.” (Lines 501-504)

L479. It would be a pertinent place to note a fact all too often forgotten by studies of these cores, that these rocks were transposed under a ~15 km thick ophiolitic nappe over 100s of km of continental margin (Searle et al. 2007) before reaching their final resting place. Rocks so close to the basal thrust could hardly escape progressive overprinting by metamorphic fluids during this translation.

This is a very good point and we have incorporated this into a more nuanced model for generation of the Stage 2 carbonating fluids:

“Fluids derived from low temperature metamorphism of autochthonous shelf carbonates during obduction are a plausible source for the F- and I-rich Stage 2 fluids, especially given the marked enrichment of Ca in the lowermost listvenites overlying the basal thrust.” (Lines 513-516)

L482: the ophiolite does not have 100 Myr subaerial exposure history, indeed, it is only 96 Ma and a large part of its life was underwater.

This was imprecise language on our part and indeed misleading; the intended meaning was ~100 Myr *near-surface* history, or words to that effect. This passage has been rephrased:

“It is likely that multiple stages of fluid infiltration affected the ophiolite prior to, during and since obduction and may have overprinted some chemical or isotopic systems.” (Lines 525-527)

L484. Last line could be more specifically geared toward carbonating fluids. There is nothing controversial about forearc slab fluid fluxing.

Agreed. Sentence has been rephrased:

“Nevertheless, our data indicate a strong subduction affinity which has escaped such overprinting and attest to fluxing and carbonation of the shallow forearc by CO₂-rich slab derived fluids.” (Line 529-531)

Line 533: Spell out IPGP.

Done

REVIEWER COMMENTS

Reviewer #3 (Remarks to the Author):

The Authors have done an exceptionally thorough job of addressing the comments, leaving tighter and more balanced/considered manuscript than previously. I congratulate them and recommend it for immediate publication.

We are delighted to read that the reviewer felt we had satisfactorily addressed their comments and thank them for their detailed and constructive criticisms which have helped to tighten up the manuscript.

Reviewer #4 (Remarks to the Author):

Carter et al. have conducted a detailed study of the mineralogy and halogen composition of ophicarbonates and listvenites from a drill core through the Oman ophiolite. The use of variation in halogen compositions in these samples to place constraints on fluid sources and the quantity of CO₂ introduced to the forearc mantle is quite novel. Both the results and the methodology used will be of interest to the subduction zone community as well as most geochemists studying volatile exchange between Earth's mantle and surface. This manuscript is well-reasoned and will be an excellent contribution.

The authors addressed previous comments from reviewers adequately. I have a few additional concerns regarding a few assumptions made by this study and the methods used.

We are pleased to read that the reviewer found our approach novel and worthy of publication and thank them for their well-considered comments which have helped clarify and strengthen several important areas of discussion.

Comments on Main Manuscript:

Lines 222-223:

The negative correlation between bulk halogen content and bulk CO₂ content is clearly demonstrated by Figure 5, but the two-stage process for Cl loss is an interpretation of the data. This could also be a single-stage process with a non-linear correlation between Cl and CO₂. The wording here needs to be clear that this is an interpretation.

This is true and indeed we did calculate an exponential fit to the data ($Cl = 0.1088\exp[-0.12399CO_2]$) and from this calculated an initial rate of Cl loss ($dCl/dCO_2 = -0.12399Cl$) even higher than that from the two-stage linear model (-1.35×10^{-2} @ $CO_2 = 0$). This was included in Table S8 for information but not further discussed in the text.

Since our calculated initial Cl loss involved projecting the exponential curve to zero CO₂, beyond our opihcarbonate data (min CO₂ = 7%) this was judged to be potentially inaccurate as the evolution of the system in the very earliest stages of carbonation (0-7% CO₂) is not well-constrained. Calculating a rate of Cl loss from the exponential curve for CO₂ = 7.06 (i.e. the least carbonated point where it is constrained) gives R_{Cl} of -5.62 x 10⁻³ very similar to the part 1 linear value of -6.43 x 10⁻³. Given the similarity of these results, we favour the linear model as being simpler. We have now clarified in the main text that the two-stage model is an interpretation and highlighted the alternate exponential model. This is further expanded upon in the supplementary text to make the above arguments explicit.

Lines 222-226: “For Cl there is rapid initial loss followed by a more gradual decrease. We interpret this as a two-stage process but the data can also be fit by an exponential curve ($Cl = a \cdot e^{-kCO_2}$) which yields a similar or higher rate of early Cl loss (Supplementary Text S1, Table S8).”

Lines 233-235: “*Change in bulk halogen abundance accompanying carbonation. (a) Cl versus CO₂ showing rapid initial Cl loss interpreted as two distinct stages of reaction with different slopes*”

Supplementary text p.3: “Alternatively, Cl loss can be fitted by exponential function as:

$$Cl = a \cdot e^{-kCO_2} \quad (10)$$

Where a is 0.1088 wt% and k = 0.12399 (with CO₂ in wt%). The rate of Cl loss against CO₂ is then itself dependent on CO₂ and given by:

$$R_{Cl} = \frac{dCl}{dCO_2} = -ka \cdot e^{-kCO_2} \quad (11)$$

$$R_{Cl} = -k \cdot Cl \quad (12)$$

[...]

Since calculating an initial rate of Cl loss from the exponential model involves projecting beyond the sample data to zero CO₂, there is a risk this may not accurately reflect the behaviour of the system in the very earliest stage of the reaction (CO₂ = 0–7 wt%). Calculated at the CO₂ content of the least carbonated sample (CO₂ = 7.06 wt%), the exponential model gives a very similar R_{Cl} to part 1 of the two-stage linear model. Given this similarity and in the interests of simplicity, we use only the linear model hereafter.”

Lines 257-259 and Figure 7c:

I don't find this claim very convincing. Serpentine Br/Cl and I/Cl from all ophiocarbonate samples seem to span most of the range of Br/Cl and I/Cl. The boxplot in Figure 7c tries to show the trend with bulk CO₂, but I don't see much of a trend in the data. However, a statistical test of correlation or differences in the Br/Cl in different samples would be the best way to show this.

Bromine/Cl measured in both bulk samples and individual SIMS spots correlates positively with bulk CO₂ for ophiocarbonate samples. A linear regression of the former gives an R² of 0.8 and p-value of 0.006 (n = 7), while the latter gives a relatively low R² of 0.16 but with a p-value of 1.4×10^{-7} (n = 165) indicates that correlation is statistically highly significant (p < 0.05) despite a large degree of scatter resulting from intrasample heterogeneity (as seen e.g. on Fig. 7a). The text has been amended to refer to these statistical tests of correlation.

Lines 261-264: “Both bulk Br/Cl and individual spots measured by SIMS show statistically significant correlations with bulk CO₂ (R² = 0.81, 0.16, respectively), the latter affected by intrasample scatter but nonetheless statistically significant (p = 1.4×10^{-7} , n = 165).”

Lines 343-355:

To use the equation on line 353 to determine the CO₂/Cl of the fluid, this paper makes assumptions about the Br/Cl of the starting fluid, the final fluid composition, and how Cl and Br change with changing CO₂ content during Stage 1 of the model. In the model presented, an initial fluid with a sedimentary pore fluid-like composition alters the peridotite forming serpentine, which then reacts with CO₂ to form carbonates, preferentially expelling Cl over Br and resulting in the expulsion of a low-Br/Cl fluid. This fluid then travels through the peridotite farther from the CO₂ reaction front, altering it and imparting the low Br/Cl signature of the fluid onto those rocks. For the Br/Cl composition of the initial fluid in the equation on line 353, a sedimentary pore fluid-like composition is used because the high-Br/Cl-I/Cl end of the sample array in Figure 7 has a composition similar to sedimentary pore fluids, correct? But wouldn't mass balance require that expulsion of a fractionated, low-Br/Cl fluid result in the partially-carbonated partially-dehydrated residue having higher Br/Cl than the initial fluid that first formed the serpentine? So, the highest Br/Cl values observed in the samples would actually be higher than the composition of the initial fluid. Of course, exactly how fractionated the Br/Cl of the residual partially carbonated samples are would depend on the extent of halogen loss, but it could be quite extensive in the most CO₂-rich samples. Unless I am misunderstanding the assumption made about the initial fluid Br/Cl composition, then this pitfall needs to be addressed.

This is a good point. However, to perform a consistent mass balance requires that all elements of the system have been preserved and this is not necessarily the case. Estimating the expelled fluid composition from the changing gradient of Cl and Br contents versus CO₂ (notwithstanding the comment below, see further answers) indicates that the expelled fluid would have higher Br/Cl in the latter stages of carbonation and indeed that the final residue would have high Br/Cl. It is possible that these fluids/solids are poorly preserved and have mostly been lost from the system. We do see a handful of very high Br/Cl points both from SIMS measurements on serpentine and NI-NG-MS on bulk listvenites (Fig. 7a). These extend to Br/Cl > 1x10⁻² and I/Cl > 4x10⁻³, consistent with a component of the mass balance which is small in volume (and consequently rarely found), but highly fractionated. A further uncertainty which prevents a full mass balance is that the reaction profile between pristine uncarbonated serpentinite and the least carbonated serpentinite is not preserved. As a result, we cannot speculate about the fractionations which may have occurred in the earliest stages of the reaction and which may have been overprinted in the rocks we see.

Perhaps more importantly, however, the large fractionations seen within the reaction zone are only possible at low water/rock ratios (which have permitted preservation of partially carbonated serpentinites; as argued in lines 381-387). At the higher water/rock ratios which characterise the formation of fully carbonated listvenites, the carbonating fluid will have been little affected by the addition of the expelled fluid. We thus expect the bulk composition of the listvenites to be close to that of the carbonating fluid (and indeed the mean listvenite composition plots close to sedimentary pore fluid).

Lastly, the pivot point for mass balance of Br/Cl should also include the contribution of any volatiles in the partially serpentinitised protolith. Since our best estimate of the composition of “normal” Oman mantle (D’Andres, 2021) has Br/Cl slightly below that of seawater or pore fluid (1.8–3.2 x 10⁻³, mean 2.4 ± 0.5; Fig. 7b) this will tend to shift the pivot point to slightly lower than the carbonating fluid.

A passage has been added to the main text to summarise the arguments made above and make the important clarification that despite fractionation within the reaction zone, elsewhere the bulk composition of the listvenites should largely reflect that of the carbonating fluid.

Lines 333-339: “Mass balance would suggest the existence of a complementary high Br/Cl component. This may be represented by a handful of serpentine data and one bulk listvenite which have very high Br/Cl and/or I/Cl, suggesting such a component is volumetrically small and potentially poorly preserved. Such a component will only form at low fluid/rock ratios however (see below) and we therefore expect the average composition of the main body of listvenite to largely reflect that of the carbonating fluid (potentially with some contribution from the uncarbonated serpentinite protolith).”

Similarly, the calculation of $(\text{Br}/\text{Cl})_{\text{out}}$ essentially determines the Br/Cl of the fluid expelled from the rock during carbonation by determining the change in bulk rock Cl and Br contents for a given change in bulk CO_2 ($\Delta\text{Cl}/\Delta\text{CO}_2$; $\Delta\text{Br}/\Delta\text{CO}_2$) and then taking the ratio of these two ratios, correct? However, the inherent assumption made by this argument and in Figure 5 is that the ophicarbonates with the lowest CO_2 and highest Cl , Br , and I contents have bulk Cl , Br , and I contents representative of the halogen contents originally in the higher CO_2 ophicarbonates and listvenites prior to their more extensive carbonation and halogen loss. However, it is later argued that the ophicarbonates with the lowest CO_2 contents have interacted with a fractionated fluid that has higher Cl and lower Br than the fluid that initially serpentinized rocks closer to the CO_2 reaction front. In that case, the Cl contents of the samples farther from the listvenites would be higher than the initial Cl contents of the samples closer to the listvenites, whereas the opposite would be true for Br . This would result in the R_{Cl} on line 347 being overestimated and R_{Br} being underestimated, which would make the calculated value of $(\text{Br}/\text{Cl})_{\text{out}}$ lower than it should be. Again, perhaps I am misunderstanding an aspect of the assumptions made, but if I am not mistaken then the potential impact of this should be addressed.

This is a very salient point. The assumptions outlined above are correct. However, the implicit assumption in our modelling which wasn't perhaps made clear enough is that we assume that a very similar profile of ophicarbonate Cl and Br existed upstream prior to further carbonation such that the listvenite front is preceded by a moving front of high Cl fluids which interact to form high Cl , low Br/Cl serpentinites. As long as such a front continues to migrate, the Br/Cl of the composition of expelled fluids will evolve as we have modelled. The implication of this is that in the earliest instance the composition of uncarbonated and unmodified serpentinites rocks and fluids likely evolved differently. But since our only evidence is the remaining rocks, we can only base our modelling on what is preserved in these. We acknowledge that the conceptual model of a steady state moving front is an assumption and have made this clear in the text. If incorrect, this could result in underestimation of the Br/Cl of the expelled fluid as suggested.

However, it can be reasoned that within the reaction zone a steady state is likely to have developed as our modelling assumes. Decoupling of rock and fluid Br/Cl vs CO_2 behaviour requires high fluid/rock ratios to supply excess of Cl to the least carbonated serpentinites. However, high fluid/rock ratios would also serve to dilute the low Br/Cl fluid from the rock increasing the fluid Br/Cl ratio from that seen. Conversely, decreasing water/rock ratios both increase the Br/Cl ratio of the fluid and tend towards equilibrium between fluids and rocks as there is not excess fluid to continually resupply halogens and CO_2 . The tendency of these countervailing effects will naturally be towards a steady state of the type we assume. Furthermore, the formation of low Br/Cl

fluid is only possible under low fluid/rock ratios suggesting that the potential for large spikes of excess fluid-derived Cl is limited. Instead, we envisage that low Br/Cl developed incrementally as a transient front at the low fluid/rock limit of carbonation. In this scenario, the CO₂/Cl constraints we calculate are those necessary for maintenance and continued propagation of this front.

It is also possible that the composition of the least carbonated serpentinites only partially reflects that of the most evolved carbonating fluid, since the mantle section was very likely partially or fully serpentinitised prior to carbonation (e.g. Beinlich et al. 2020, JGR: SE). Given this, the composition of the low Br/Cl endmember represents a maximum fluid composition and might represent an intermediate composition between the serpentinite protolith and a more fractionated (lower Br/Cl) fluid. Carried through to calculating CO₂/Cl of the carbonating fluid the effect of this would tend to cancel out any potential overestimation of Br/Cl_{out} resulting from the considerations above suggesting the modelling represents a reasonable estimate based on the available constraints.

The text has been modified to clearly indicate this important assumption the potential impact of any divergence from this. Further justification of this assumption including the arguments above has been included at the start of Supplementary Text S1 preceding the derivation of modelling equations.

Lines 368-374: “Since the least carbonated serpentinites have been affected by interaction with fractionated, Cl-rich fluid, this approach assumes that the compositional gradients across the reaction zone had reached a steady state (i.e. less carbonated samples evolve to resemble more carbonated samples; see Supplementary Text S1). If this is not the case it is possible that the Br/Cl_{out} calculated is an underestimate and the initial CO₂/Cl of the carbonating fluid required would be even higher.”

Supplementary text p.1: “It should be noted that the least carbonated ophicarbonates have been affected by interaction with fractionated low Br/Cl fluids. A key assumption in the following calculations is therefore that compositional gradients of halogens and CO₂ content across the reaction zone (main text Figure 5) were in steady state, such that all samples analysed evolved (or would have evolved) along the same compositional gradient. This is consistent with the view of the reaction zone as a transient compositional front and implies that it migrated with an unchanging compositional profile with respect to halogens and CO₂. If this is not the case, then the rate of Cl loss during carbonation may be overestimated relative to other halogens, resulting in underestimation of CO₂/Cl through the following calculations. That said, it is reasonable to think that such a steady state should have evolved since low Br/Cl fluids can only be maintained at low fluid/rock ratios, limiting their ability to supply excess Cl to

uncarbonated rocks downstream. Conversely, increasing fluid/rock ratios increases the capacity of the fluids to deliver excess Cl downstream rocks but at the same time increases the Br/Cl of those fluids. The existence of low Br/Cl serpentinites affected by a fractionated fluid therefore suggests a balance between these countervailing tendencies. It is also possible that the least carbonated ophiocarbonates are compositionally intermediate between their serpentinite protolith and the evolved carbonating fluid, since they were most likely serpentinitised prior to carbonation¹. The effect of this would tend to cancel out any potential overestimation of Br/Cl_{out} and suggests that these assumptions and the modelling approach which follow are both reasonable given the available constraints.”

Figure 6:

There are error bars with no associated data in both a) and b).

There are a few samples with larger than normal errors (for most data points they are smaller than the symbol). These appear asymmetric due to the logarithmic scale used. The caption has been amended to make this clear and to state that the plot is log-log.

Lines 279-281: “Error bars are smaller than the symbols for most data points and, where visible, appear as asymmetric due to the logarithmic scale.”

Comments on Methods Section:

Lines 604-605:

There are many potential problems with measuring halogens by pyrohydrolysis-IC-ICP-MS that need to be clear to the reader. As an example, if the calibration standard solutions are not appropriately matrix matched to the same NaOH concentration as the samples, there can be significant matrix effects on the apparent Br content of the samples. There is a reference to Balcone-Boissard et al., (2009) for the method, but there needs to be some additional detail given here. Not all the details of the method need to be added, but I would like to know at least some details of the ion chromatography and ICP-MS methods (e.g., what standard concentrations were used, were they matrix matched to the standards for ICP-MS).

Analyses were made using calibration curves across the range of halogen concentrations in sample/rock standard solutions and for ICP-MS were matrix matched to avoid potential bias. These and several other important details have been added to the description of the methods.

Lines 633-649: “Both methods were conducted with standard calibration curves bracketing the range of halogen contents in sample and rock standard solution. For IC these were prepared by dilution of commercial standard in deionised water with 2% v/v HNO₃. For ICP-MS the calibration solutions were likewise prepared from commercial standards and matrix matched to the final solutions from pyrohydrolysis (5% v/v NaOH,

2% v/v HNO₃) to avoid any potential matrix effects⁶⁹. In both cases calibrations were linear for all elements analysed. Repeat analyses of rock standards (BHVO-2, GS-N, JG-1, JR-1, JB-1b, JB-2) indicate mean relative errors of ~10% for F and Cl, ~30% for Br and ~60% for I, and yields of >95% for all halogens⁶⁹. Detection limits in solution were well below sample concentrations for F and Cl (<100 µg/L) and were approximately 10ppb and 1ppb for Br and I, respectively. For extraction of 500 mg of sample, producing 100 ml of pyrohydrolysis solution, detection limits in rock samples are estimated to be 20 mg kg⁻¹ for F and Cl and 2 µg kg⁻¹ for Br and I. The mean analytical errors on the studied composition ranges were estimated at 10 mg kg⁻¹ for F and Cl, 100 µg kg⁻¹ for Br and 25 µg kg⁻¹ for I. No blank correction was necessary as procedural blanks (both tube and tube + platinum crucible) were always below detection limits. The batch of V₂O₅ used was tested prior to analysis to ensure it did not contain detectable halogens.”

Line 606:

Was the V₂O₅ heat treated prior to fusion? I know that Balcone-Boissard et al. (2009) indicated that the V₂O₅ did not contain significant quantities of halogens, but Segee-Wright et al. (2023) indicated that different V₂O₅ batches (even with the same product number and from the same manufacturer) can have starkly different halogen contents, particularly for Br and I. Unless this study used the exact same batch of V₂O₅ used by Balcone-Boissard et al. (2009), then there needs to be some concern about contamination from the V₂O₅. Line 617 mentions that procedural blanks were run. Did those procedural blanks include V₂O₅? This is something that can be quickly addressed.

Each batch of V₂O₅ accepted into the lab, including that used for these analyses, was checked for purity in the same manner as described by Balcone-Boissard et al. (2009) and no detectable halogens were found. Procedural blanks included both tube and tube + Pt crucible. The method has been amended to include these details.

See **lines 633-649** above.

Lines 613-614:

Here, the names of the rock standards measured need to be reported, as well as the calculated yields are for each halogen. Without this information, it is challenging to assess the quality of this data.

BHVO-2, GS-N, JG-1, JR-1, JB-1b, JB-2 were run as standards and indicated yields of >95% for all halogens. This information has been added to the method.

See **lines 633-649** above.

Line 615:

Is this supposed to say <100 g/L? Because the is equivalent to 100,000 µg/mL or 10 wt%

F or Cl, which is impossibly high for 0.5 grams of fluxed rock at ~80 mL of solution extracted. I assume this is a typo of some sort. What was this supposed to say?

Yes indeed this was a typo and should have read <100 µg/L. The text has been amended.

See **lines 633-649** above.

Line 616:

Is this in the solution or propagated to the rock? Either way, both should be reported for every halogen.

These were for solution analysis. This has been clarified and the corresponding detection limits and analytical errors in rock are now included in the methods.

See **lines 633-649** above.

Line 684:

^{18}O and ^{30}Si were measured on an electron multiplier in serpentines that have high O and Si contents? Wouldn't the counts be extremely high and damage the EM over time? Faraday cups weren't used for measuring those elements? I'm not saying this is wrong, I'm just checking to make sure this is correct because it is different from the volatile measurement procedures I've used at other SIMS facilities.

Having checked the original condition files, both ^{18}O and ^{30}Si were indeed measured on the electron multiplier.

Lines 689-691:

Show the good correlation between EPMA and SIMS Cl measurements in a supplementary figure.

This is now included in the supplementary text along with figures demonstrating the linearity of SIMS signals for Cl and Br over a range of standard matrices and halogen concentrations.

Supplementary text p.10

Lines 690-692:

Was only one standard used to make F and Cl calibration curves? I don't like the idea of using only one calibration standard for Cl and F, particularly when the standard is not matrix matched. This is a problem for both F and Cl, but at least the F contents of most samples are lower than that of the standard, so their values can be interpolated between the standard and the origin. For Cl, the standard has 113 ppm Cl, but the samples extend to 900-1100 ppm Cl. The error associated with extrapolating a calibration curve to 9 times higher than the one standard will result in enormous errors. Do the errors reported for the SIMS data take this extrapolation into account? If they don't, they need to. Why wasn't a higher Cl standard chosen?

A single primary standard was indeed used for both F and Cl. In choosing this we were led by the judgement of the staff at the Edinburgh Ion Microprobe Facility where the analyses were conducted. Uncertainty in the relative ion yield determined from standards will scale linearly with increasing sample Cl/O and such uncertainties are already propagated into the errors quoted for Cl etc. If the sensitivity of the SIMS for Cl deviated from linear this could indeed impart large systematic errors in the determination of Cl extrapolated beyond the standard values. However, two lines of evidence suggest this is not the case.

Firstly, the rhyolitic glass ATHO was also run during method set up. Processing T1G data using ATHO as the primary standard and *vice versa* for ATHO gives F and Cl abundances that are within 10% of the accepted value for each standard and thus within typical propagated errors, despite ATHO having 3 and 4 times higher F and Cl, respectively, than T1G. These data are now provided as Supplementary Table S9 and discussed in the methods.

Further demonstration of the validity of the data for Cl is provided by the strong linear correlation between Cl abundance measured by SIMS and EPMA (see above comments) with a slope of 1.04 and R^2 of 0.81. The data show no indication of systematic deviation from this trend to suggest any non-linearity in SIMS signal at high Cl contents. We also note that the correlation between our Cl datasets appears highly comparable or indeed somewhat better than similar data available in the literature (e.g. Cassidy et al. 2022, *Am. Mineral.* 107, supp. material). Nonetheless we recognise that there may be some unaccounted-for uncertainty in the highest Cl values and have added discussion of the above evidence for linearity and a cautionary note to the methods section in this regard.

Lines 727-740: “It should be noted that some serpentine analyses had Cl significantly above that of the primary standard. If signal response was not linear with increasing Cl this could result in large systematic errors. However, linearity of the signal was assessed by analysis of two natural glasses with different SiO_2 contents and Cl contents (Supplementary Figs. S1-2; T1G, 58.6 wt% SiO_2 , ATHO, 75.6 wt% SiO_2). Processing data from each standard using the other as the primary standard gives F and Cl abundances that are within 10% of the accepted value for each. The standard deviation on the calculated relative ion yields was propagated together with the analytical error on each point and includes the increase in uncertainty involved with extrapolating beyond the calibration range. Moreover, the SIMS Cl data can be checked against EPMA data from the same points and show a strong linear correlation with a slope of 1.04 (Supplementary Fig. S3). The combined evidence therefore points to the reliability of these data. Nonetheless, the potential for unaccounted-for systematic error in the highest Cl analyses should be noted.”

Lines 704-707:

Instead of telling the reader that there is a linear relationship, show this in a supplementary figure. This is especially important since these measurements are not routine. If scapolites and glasses of different compositions also fall along a single calibration line, that could be used as an argument against significant matrix effects for these measurements.

These data indeed show a single calibration line for scapolites and synthetic glasses over a wide range of Br contents. A plot of these data is now included in the Supplementary Text as Fig. S4. We also note that a very similar and linear Br SIMS response was reported by Cassidy et al. (2022, *Am. Mineralogist*) and they similarly argued that matrix effects were likely to be minimal. The linearity of signals in this and the study of Cassidy et al. are now referenced as evidence against significant matrix effects as suggested.

Lines 775-779: “Nonetheless, the good agreement of SIMS, EPMA and NI-NG-MS data (see below, Supplementary Fig. S3) lends confidence that any matrix effects are

relatively minor as do the linear responses of F, Cl and Br across a range of standard matrices in this study (Supplementary Figs. S1-2, S4) and others in the literature⁷⁶

Supplementary Text p8-11:

Lines 729-737:

Although I do not like that there are no matrix matched standards or that only one calibration standard was run, the reasoning in the section is decent. I don't think the problems I have raised regarding the methods will have a large impact on the main conclusions of this paper.

However, the claim that matrix effects are minimal when comparing two different elements within the same mineral needs to be justified, either through a literature reference or by making the argument in this section. This could be demonstrated by comparing SIMS calibrations of two different elements (F and Cl for example, but it could really be any two elements) in two sets of standard materials with very different matrices that have been previously published. If the ratio of the slope of the Cl calibration line to the slope of the F calibration line is similar for both standard sets, then one could argue that the matrix will have no significant impact on the measured F/Cl ratio.

The reliability of the F/Cl ratio is demonstrated by the strong linearity of F and Cl signals across a wide range of glass SiO₂ contents. The ratios of the slope of the calibration lines for F and Cl (i.e. RIY_F/RIY_{Cl}) for both standards are within 20% of each other. This strongly suggest that the large variation seen within minerals species (carbonate F/Cl = 0.02 to 398, serpentine F/Cl = 0.02 to 1.48) cannot be explained by matrix effects. Further argument against an important role for matrix effects is the occurrence of vein margin magnesite with high F/Cl (<108) and a few grains of dolomite with low F/Cl ratios (<1; see Fig. 6). The key distinction controlling the F/Cl ratio measured appears to be textural rather than mineralogical suggesting it is indeed geological factors not matrix effect that gives rise to these distinctions. A passage has been added to the method section to make this line of argument explicit.

Lines 782-790: “The ratio of F and Cl relative ion yields for two natural glasses with a wide range of SiO₂ contents are within 20% of one another suggesting matrix effects cannot explain the large variations seen in data (carbonate F/Cl = 0.02 to 398. serpentine F/Cl = 0.02 to 1.48). Furthermore, within carbonates, the principal control on F/Cl appears to be textural rather than mineralogical with interstitial dolomite and magnesite both showing high F/Cl (Fig. 6b). Nonetheless, it is impossible to discount that some matrix effects may affect the data and particularly absolute concentrations. As a result, care should be taken in making direct comparison of concentrations between different minerals.”

See also Supplementary plots above

Even if matrix effects don't change F/Cl, Br/Cl, or I/Cl values by orders of magnitude, matrix effects will almost certainly have some systematic effect on the absolute concentrations of all halogens that will differ between different phases. Although the

halogen ratios are unlikely to change enough to alter the interpretations of the manuscript, the concentrations could be systematically offset by >50-100%. There needs to be mention of the deficiencies of the method and the resulting deficiencies of the data in respect to absolute halogen concentrations.

We acknowledge this is a fair point and have added a note to the methods cautioning against direct comparisons of concentrations between samples.

See **Lines 782-790** above

REVIEWERS' COMMENTS

Reviewer #4 (Remarks to the Author):

The authors have adequately addressed my concerns in the revised manuscript and supplement. I recommend that this manuscript for publication.

We are very pleased to read that our revisions addressed the concerns of the reviewer and that they now recommend it for publication.